# Dynamic multicolor emissions of multimodal phosphors by Mn$^{2+}$ trace doping in self-activated CaGa$_4$O$_7$

Yiqian Tang[1,2,6], Yiyu Cai[1,2,6], Kunpeng Dou[1,2], Jianqing Chang[1,2], Wei Li[1,2], Shanshan Wang[1,2], Mingzi Sun [3], Bolong Huang [3] ✉, Xiaofeng Liu[4], Jianrong Qiu [4], Lei Zhou[5], Mingmei Wu[5] & Jun-Cheng Zhang [1,2] ✉

The manipulation of excitation modes and resultant emission colors in luminescent materials holds pivotal importance for encrypting information in anti-counterfeiting applications. Despite considerable achievements in multimodal and multicolor luminescent materials, existing options generally suffer from static monocolor emission under fixed external stimulation, rendering them vulnerability to replication. Achieving dynamic multimodal luminescence within a single material presents a promising yet challenging solution. Here, we report the development of a phosphor exhibiting dynamic multicolor photoluminescence (PL) and photo-thermo-mechanically responsive multimodal emissions through the incorporation of trace Mn$^{2+}$ ions into a self-activated CaGa$_4$O$_7$ host. The resulting phosphor offers adjustable emission-color changing rates, controllable via re-excitation intervals and photoexcitation powers. Additionally, it demonstrates temperature-induced color reversal and anti-thermal-quenched emission, alongside reproducible elastic mechanoluminescence (ML) characterized by high mechanical durability. Theoretical calculations elucidate electron transfer pathways dominated by intrinsic interstitial defects and vacancies for dynamic multicolor emission. Mn$^{2+}$ dopants serve a dual role in stabilizing nearby defects and introducing additional defect levels, enabling flexible multi-responsive luminescence. This developed phosphor facilitates evolutionary color/pattern displays in both temporal and spatial dimensions using readily available tools, offering significant promise for dynamic anticounterfeiting displays and multimode sensing applications.

Counterfeiting is a critical global issue that poses significant risks to public health, safety, and economic stability across losses in various industries[1,2]. Luminescent materials have gained increasing popularity in the field of anti-counterfeiting techniques due to their high sensitivity to external stimuli, ease of visual observation, and compatibility with existing manufacturing processes for encryption labels[3–6]. The excitation mode and resulting emission color of luminescent materials

are pivotal factors in encrypting information for anti-counterfeiting purposes[7–11]. To enhance the security level of anti-counterfeit labels, traditional luminescent materials with a monomodal emission are being upgraded to single-component luminescent materials with multimodal emissions. The developed multimodal luminescent materials, including coordination polymers[12,13], carbon dots[14,15], nanocrystals[16,17], and lanthanide doped inorganic phosphors[18,19], can

emit tunable colors by adjusting the type or parameters of the external stimulus, such as ultraviolet (UV)/near-infrared photoexcitation, X-ray irradiation, thermal disturbance, or mechanical stimulation. These single-component luminescent materials provide rich spatial recognition information for encrypted displays and overcome the complexities and inefficiencies associated with composite structures and mixed materials[20–22]. Nevertheless, a limitation of current multimodal luminescent materials is their static monochromatic luminescence, wherein the emission color remains unchanged under a fixed excitation. This lack of temporal encryption hinders the achievement of high-level anti-counterfeiting in both spatial and temporal dimensions.

Recent developments in dynamic multicolor luminescent materials introduce a temporal layer of authentication for emission colors[23]. Dynamic multicolor luminescence is typically achieved through two emission bands of different colors that exhibit distinct time-dependencies in accumulation or decay rates, resulting in color evolution over time in the overall luminescence[12,24]. While most current materials with dynamic multicolor luminescence rely on phosphorescence or afterglow after stopping light excitation[25,26], these emissions are generally weak and rapidly decay, making them unsuitable for visually recognizing dynamic color evolution. A more effective solution lies in PL materials with time-dependent multicolor changes[27,28]. These materials consistently exhibit intense emission under a fixed light excitation, enhancing color discernibility by the human eye. However, reports on dynamic multicolor PL materials are extremely limited, and none of them exhibit multimodal response[24,27,29–31]. Achieving dynamically varying multicolor emission with multimodal response within a single material remains a daunting challenge.

Defect-controlled inorganic phosphors offer significant potential for the development of dynamic multimodal phosphors. The presence of diverse defect levels resulting from intrinsic or extrinsic defects, along with their intricate interactions with luminescent levels, creates numerous opportunities to construct advanced luminescent properties within a single material[32,33]. For example, multimodal phosphors have been developed by incorporating dual-lanthanide ions into non-luminescent lattices[34,35]. This approach enables the coupling of trapping and de-trapping processes and optical multiplexing of dual-lanthanide dopants, resulting in phosphors with distinct responses to various physical fields induced by external photo-thermal-mechanical stimuli. Moreover, defect

levels can modulate the energy distribution of the excitation states through tunneling effects or energy transfer, leading to different kinetic processes for each emission band and the production of dynamic multicolor emission[24,31]. Despite significant efforts, several aspects regarding the nature of intrinsic and extrinsic defects, the trapping and de-trapping pathways under different external stimuli, and the kinetics of dynamic multicolor emission governed by versatile defects remain unclear. These knowledge gaps impede the development of a single material with dynamic multimodal luminescence.

In this study, we propose a strategy to achieve dynamic multimodal luminescence by incorporating trace amounts of transition metal ions $Mn^{2+}$ into the self-activated gallate host $CaGa_4O_7$ (Fig. 1). The gallate host was selected for its exceptional properties, including self-activated luminescence[36,37], a lattice suitable for activating transition metal ions[38,39], an electronic structure conducive to trap-controlled luminescence[40,41], and high physicochemical stability[42]. The combination of the host's blue emission with the yellow emission from $Mn^{2+}$ ions fulfills the necessary spectral criteria for achieving dynamic multicolor luminescence. The incorporation of $Mn^{2+}$ ions also creates extrinsic defects within the host, working synergistically with intrinsic defects to establish multiple defect levels, thus facilitating the generation of the desired dynamic multimodal luminescence. The resulting $CaGa_4O_7$:$Mn^{2+}$ phosphor exhibits time-dependent dynamic multicolor luminescence (ranging from yellow to brown to cool white) under fixed photoexcitation, featuring a temporal memory effect on the re-excitation intervals. Additionally, the phosphor shows temperature-dependent color reversal with anti-thermal-quenched $Mn^{2+}$ emission. Furthermore, it displays intense elastic ML with high mechanical durability. Each of these luminescent features is uncommon, and their integration within a single material is inaccessible by any previously reported luminescent materials (Supplementary Table 1). Density functional theory (DFT) calculations systematically demonstrate the intrinsic defect levels in both host material and $CaGa_4O_7$:$Mn^{2+}$, revealing the potential electron transfer mechanisms for the dynamic multicolor emission and multi-responsive luminescence. In particular, the vacancies and interstitial defects play a significant role in realizing the dynamic multimodal luminescence. Finally, by leveraging commonly used tools such as UV lamps, heating stimuli and pen writing, a multilevel luminescent anti-counterfeiting demonstration is designed, showcasing the phosphor's ability to sense photo-thermal-mechanical stimuli and display dynamic evolution of colors and patterns in the temporal and spatial dimensions.

## Results

### Controlled incorporation of $Mn^{2+}$ ions

In the proof-of-concept experiment, $Ca_{1-x}Ga_4O_7$:$xMn^{2+}$ ($x = 0–10 \times 10^{-4}$) phosphors were synthesized through a solid-state reaction involving $CaCO_3$, $Ga_2O_3$ and $MnCl_2$. X-ray diffraction (XRD) analysis verifies the crystallization of the resulting phosphors as a single phase (Supplementary Fig. 1). The PL properties of these phosphors were systematically investigated, including PL spectra, decay curves and quantum yields (PLQY) (Supplementary Figs. 2–4). The steady-state PL spectra show that the luminescence of the $CaGa_4O_7$:$Mn^{2+}$ series consists of blue emission from the self-activated $CaGa_4O_7$ host and yellow emission from the doped $Mn^{2+}$ ions (Supplementary Fig. 2). Notably, the blue emission gradually decreases as the $Mn^{2+}$ concentration increases, whereas the yellow emission increases gradually. This opposite tendency in emission allows the trace $Mn^{2+}$ ($x = 1 \times 10^{-4}$) doped phosphor to achieve an optimal blue-yellow emission ratio, facilitating the generation of multicolor emission. Therefore, the following discussion focuses on the phosphor $CaGa_4O_7$:$Mn^{2+}$ with $x = 1 \times 10^{-4}$.

### Microstructural characterization

Figure 2 illustrates the microstructural characterization of the $CaGa_4O_7$:$Mn^{2+}$ phosphor. High-resolution transmission electron

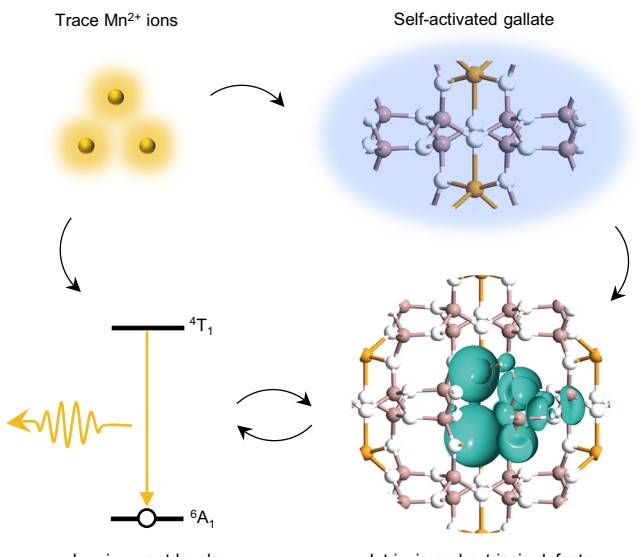

Trace $Mn^{2+}$ ions

Self-activated gallate

$^4T_1$

$^6A_1$

Luminescent levels

Intrinsic and extrinsic defects

**Fig. 1 | A strategic approach for achieving dynamic multimodal luminescence within a single material.** This strategy involves incorporating trace $Mn^{2+}$ ions into a self-activated gallate host, fostering interactions between intrinsic and extrinsic defects, as well as the luminescent levels of $Mn^{2+}$ dopants.

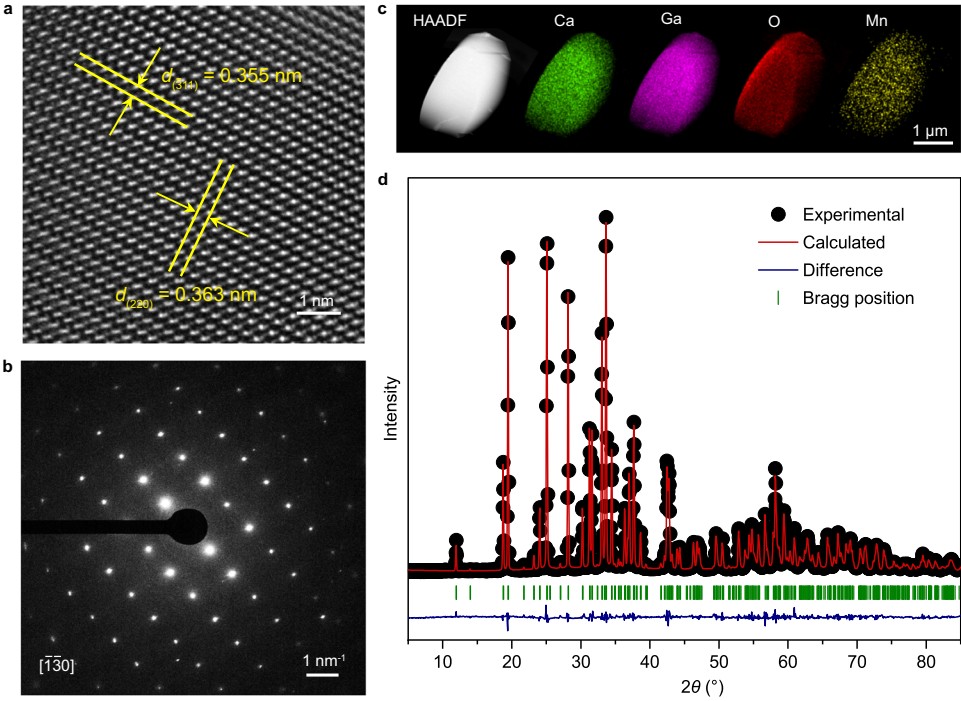

**Fig. 2 | Structural characterizations of as-synthesized CaGa$_4$O$_7$:Mn$^{2+}$. a** HRTEM image displaying lattice fringes in *d*-spacing. The yellow lines show the corresponding *hkl* planes and the yellow arrows indicate the lattice spacing. **b** SAED pattern presenting the view down of the [$\bar{1}$30] projection. **c** HAADF-STEM image and corresponding compositional analysis of an individual particle. **d** XRD pattern with the calculated profile obtained from Rietveld refinement. Source data are provided as a Source Data file.

microscope (HRTEM) images and selected-area electron diffraction (SAED) patterns reveal the single-crystal nature of individual particles (Fig. 2a, b). The SAED pattern exhibits lattice fringes characterized by *d*-spacings of 0.355 and 0.363 nm (inserted in Fig. 2a), corresponding to the ($\bar{3}$11) and (220) facets along the [$\bar{1}$30] direction of monoclinic CaGa$_4$O$_7$, respectively. Compositional analysis conducted through high-angle annular dark-field scanning transmission electron microscope (HAADF-STEM) confirms the uniform distribution of all target elements (Ca, Ga, O and Mn) within the structure of an individual particle (Fig. 2c). Rietveld refinement of the crystal structure provides additional verification that the synthesized CaGa$_4$O$_7$:Mn$^{2+}$ particles exhibit single-phase crystals with a monoclinic structure of space group *C*2/*c* (No. 15) (Fig. 2d and Supplementary Table 2). The similar cationic sizes of Ca$^{2+}$ and Mn$^{2+}$ ions, along with the crystal field effect of the five-coordinated calcium, facilitate the preferential occupation of the Ca$^{2+}$ sites by Mn$^{2+}$ dopants, resulting in the yellow emission (Supplementary Table 3 and Supplementary Fig. 5).

**Dynamic multicolor PL**
The CaGa$_4$O$_7$:Mn$^{2+}$ phosphor exhibits dynamic multicolor PL at room temperature (Fig. 3). Under continuous photoexcitation with a power density of 0.50 W m$^{-2}$ from a handheld 254 nm lamp, the emission color of the phosphor spontaneously transitions from yellow to brown to cool white within 10 s (Fig. 3a). Time-dependent PL spectra reveal that the blue emission of the CaGa$_4$O$_7$ host (peaking at 474 nm) gradually increases with longer photoexcitation time, while the yellow emission of Mn$^{2+}$ (peaking at 579 nm) gradually decreases (Fig. 3b and Supplementary Fig. 6). This reverse variation in blue and yellow emission intensities leads to dynamic multicolor changes during photoexcitation. Photoluminescence excitation (PLE) spectra, monitored at different emission wavelengths ranging from 400 to 625 nm, exhibit identical shapes and features, with an excitation peak observed at 256 nm (Fig. 3c and Supplementary Fig. 7). Diffuse reflectance spectra demonstrate that the fixed excitation peak

correlates with the host's absorption band (Supplementary Fig. 8), suggesting that the excitation processes for both host emission and Mn$^{2+}$ emission in CaGa$_4$O$_7$:Mn$^{2+}$ are dominated by the transition from the valence band to the conduction band[43]. Control experiments conducted on undoped CaGa$_4$O$_7$ material confirm that the enhanced blue emission with increasing photoexcitation time is an intrinsic property of the CaGa$_4$O$_7$ host (Fig. 3d, e and Supplementary Fig. 9). The kinetic process of PL shows that the blue emission reaches near saturation after 7 s of UV irradiation (Fig. 3f). It suggests that the blue emission is associated with the accumulation of excited electrons in the defect states[24,44].

**Adjustable emission color-changing rates**
We observed an intriguing phenomenon that the emission color-changing rate of CaGa$_4$O$_7$:Mn$^{2+}$ exhibits a temporal memory effect on the re-excitation interval. A shorter time interval between two excitations leads to a faster multicolor evolution. Specifically, upon a fixed optical excitation (254 nm, 0.50 W m$^{-2}$), the evolution time from yellow to cool white is approximately 4.5 s, 6 s and 8 s at re-excitation intervals of 60 s, 300 s and 900 s, respectively (Fig. 3g). Time-resolved PL spectra of CaGa$_4$O$_7$:Mn$^{2+}$ reveal that the initial intensity of the yellow emission decreases, and the decay rate slows down as the re-excitation interval shortens (Fig. 3h and Supplementary Figs. 10 and 11). In contrast, the re-excitation interval has a limited effect on the blue emission (Fig. 3h). Therefore, the strong dependence of the yellow emission on the re-excitation interval is the primary reason for the observed temporal memory effect. Interestingly, a simple heat treatment (e.g., 473 K for 10 s) can rapidly restore the initial evolutionary rate when the phosphor is first photoexcited (Fig. 3g). Under cyclic photoexcitation and heat treatment, the dynamic multicolor evolution demonstrates excellent reproducibility and rapid operability (Supplementary Fig. 12). The results suggest that the dynamic multicolor evolution of CaGa$_4$O$_7$:Mn$^{2+}$ is closely related to trap-mediated trapping and de-trapping processes, which will be discussed in detail later.

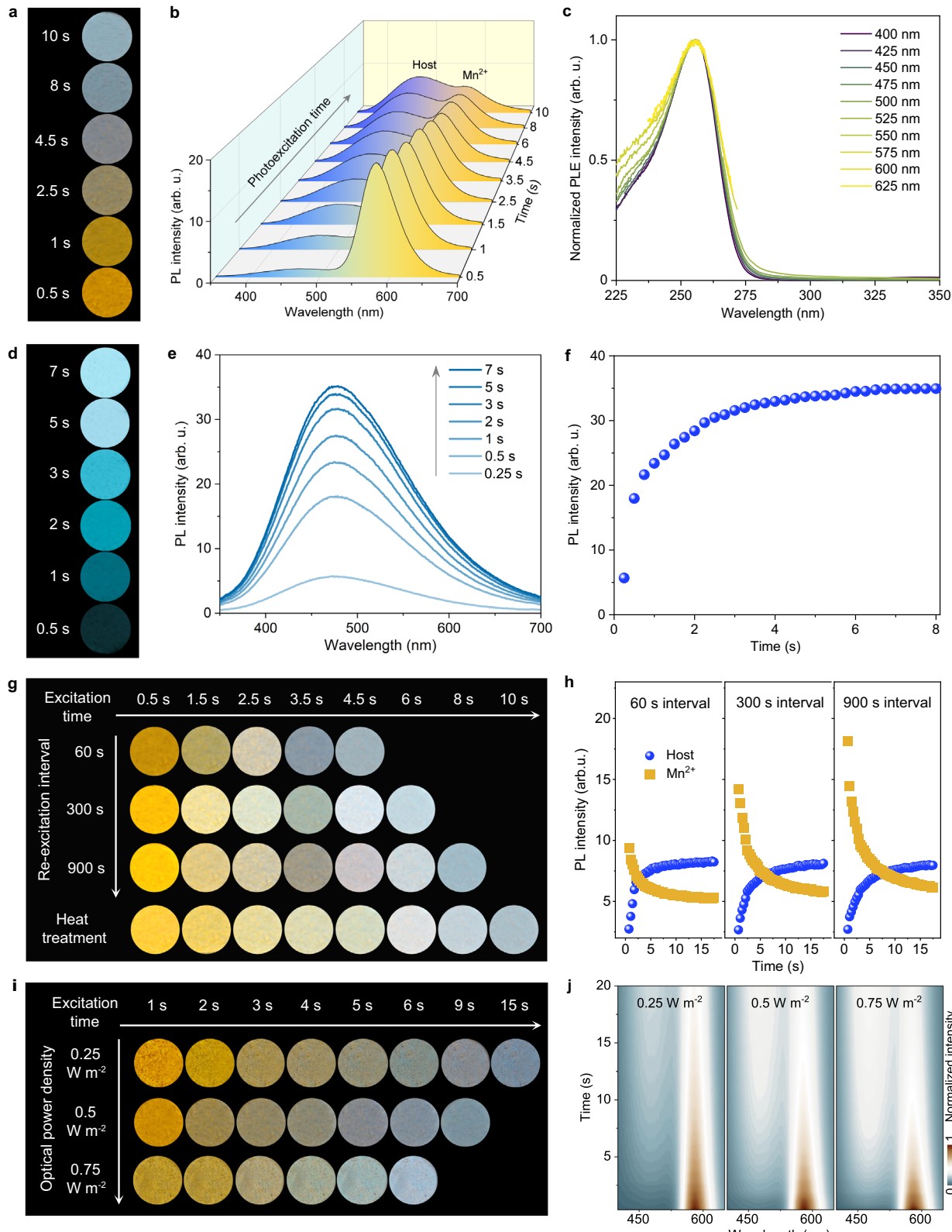

**Fig. 3 | Dynamic multicolor evolution of PL at room temperature.**
**a**, **b** Photographs and spectra of PL from $CaGa_4O_7$:$Mn^{2+}$ with increasing photoexcitation time. **c** Excitation spectra of $CaGa_4O_7$:$Mn^{2+}$ monitoring different emission wavelengths. **d**–**f** Photographs, spectra, and kinetic process of PL from undoped $CaGa_4O_7$ with increasing photoexcitation time. **g** Photoexcitation time-dependent PL photographs of $CaGa_4O_7$:$Mn^{2+}$ at different re-excitation intervals and after heat treatment. **h** Kinetic processes of blue and yellow emissions re-excited after different time intervals. **i**, **j** Time-dependent PL photographs and transient PL images of $CaGa_4O_7$:$Mn^{2+}$ under different optical power densities. Source data are provided as a Source Data file.

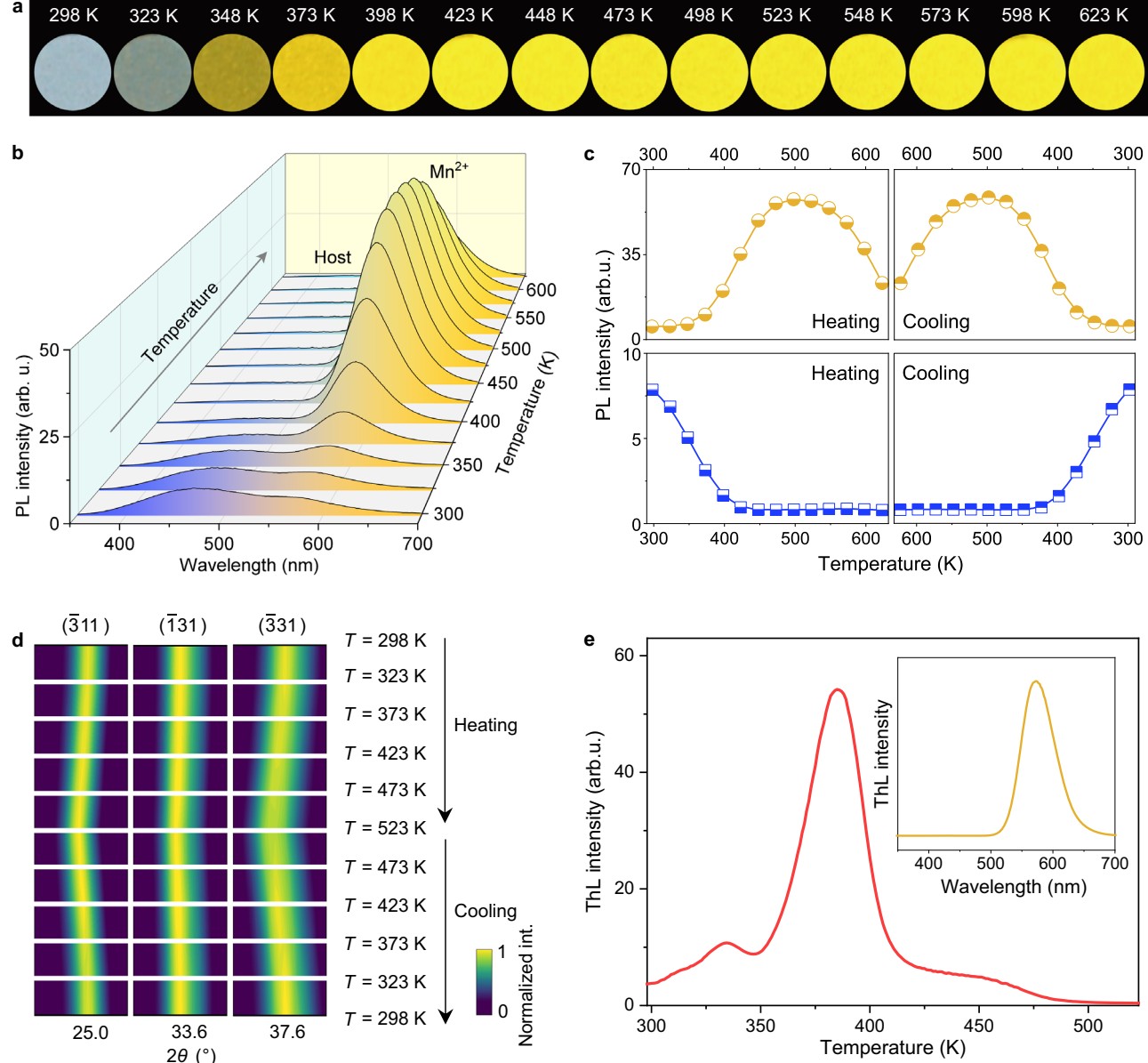

**Fig. 4 | Temperature-dependent multicolor luminescence and thermally boosted yellow emission. a, b** Photographs and spectra of PL from $CaGa_4O_7$:$Mn^{2+}$ heated from 298 K to 623 K. **c** Relative PL intensity of blue emission (blue squares) and yellow emission (yellow cycles) during the heating (298 – 623 K) and cooling (623 – 298 K) cycle. **d** Representative XRD patterns of $CaGa_4O_7$:$Mn^{2+}$ heated from 298 K to 523 K and subsequently cooled to 298 K. **e** ThL curve of $CaGa_4O_7$:$Mn^{2+}$. The inset shows the ThL spectrum collected during heating. Source data are provided as a Source Data file.

Another interesting finding is that the change rate of emission color from $CaGa_4O_7$:$Mn^{2+}$ can be modulated by the power density of the excitation light. Increasing the power density of the excitation light accelerates the evolution of the emission color, with evolution times ranging from approximately 15 s at 0.25 W m$^{-2}$ to around 9 s at 0.50 W m$^{-2}$ and approximately 6 s at 0.75 W m$^{-2}$ (Fig. 3i and Supplementary Fig. 13). The transient PL images show that as the optical power density increases, the locations of the blue and yellow emissions remain constant, but the enhancement rate of the blue emission and the decay rate of the yellow emission become faster, thus accelerating the evolution of the overall emission color (Fig. 3j). Notably, the ability to control the change rate of emission color by adjusting both the power density of the excitation light and the re-excitation interval can be harnessed to encode distinguishable temporal fingerprints. This characteristic ensures easy authentication while presenting challenges for replication in practical anti-counterfeiting applications.

## Temperature-dependent multicolor emissions

The $CaGa_4O_7$:$Mn^{2+}$ phosphor exhibits temperature-dependent multicolor PL behavior (Supplementary Fig. 14), accompanied by thermally boosted $Mn^{2+}$ emission (Fig. 4). As the temperature increases from 298 to 373 K, the emission color reverses from cool white to yellow, and with further temperature increases, the yellow emission becomes even brighter. Even at a high temperature of 623 K, the yellow emission remains more intense than that observed at 373 K (Fig. 4a). The temperature-dependent PL spectra reveal that the blue emission from the $CaGa_4O_7$ host is subject to a common thermal quenching (Fig. 4b and Supplementary Fig. 15). In contrast, the yellow emission from $Mn^{2+}$ experiences an unusual thermal enhancement within the temperature range of 298 – 498 K, exhibiting an impressive tenfold enhancement. Although a portion of the enhanced yellow emission weakens with further temperature increases, the relative luminous intensity at 623 K is still 4.4 times greater than that at 298 K (Fig. 4b, c). Notably, the

changes in blue and yellow emissions during heating can be completely reversed by cooling, highlighting excellent temperature dependence and reproducibility (Fig. 4c). This combination of temperature-dependent color change and thermally boosted emission is an emission feature rarely observed in reported luminescent materials, making it well suited for robust anti-counterfeiting applications with high levels of security associated with thermal stimulation.

To investigate the possibility of phase transitions during temperature variations, XRD measurements of $CaGa_4O_7:Mn^{2+}$ were conducted at different temperatures. The results clearly indicate that there was no crystalline phase transition throughout the heating-cooling cycle (Supplementary Fig. 16). Slight shifts in the characteristic diffraction peaks were observed, which can be attributed to the thermal expansion and contraction of the lattice (Fig. 4d). Additional spectral observations demonstrate that the location of the yellow emission shifts towards longer wavelengths with heating and shorter wavelengths with cooling, further supporting the occurrence of thermally induced lattice deformation without a transition in the crystal structure (Supplementary Fig. 17). Subsequently, thermoluminescence (ThL) tests were performed to investigate the correlation between traps and luminescence. The ThL curves exhibit three distinct peaks within the temperature range of 298–523 K for the $CaGa_4O_7:Mn^{2+}$ phosphor (Fig. 4e), indicating the presence of three traps with estimated depths of 0.59, 0.68, and 2.34 eV (Supplementary Fig. 18)[45]. Importantly, the temperature range of the three ThL peaks (Fig. 4e) corresponds to the temperature range of the thermally boosted yellow emission (Fig. 4c), both falling within the 298–498 K range. Furthermore, the ThL spectrum collected during heating reveals a single emission peak corresponding to the yellow emission of $Mn^{2+}$ ions, while no blue emission from the host was detected (inset of Fig. 4e), indicating a connection between the traps and $Mn^{2+}$ ions. Based on these ThL results, we suggest that the thermally boosted yellow emission originates from the energy replenishment from the traps to the $Mn^{2+}$ ions, which outweighs the energy loss caused by the non-radiative multiphonon transition at high temperature[46–48]. In contrast, the blue emission undergoes thermal quenching due to the lack of energy supply from the traps.

## Durable elastic ML

The incorporation of the $CaGa_4O_7:Mn^{2+}$ particles into optical polymer elastomers enables the nondestructive manifestation of ML properties when subjected to mechanical stimuli (Fig. 5a). Applying mechanical friction to pre-photoexcited $CaGa_4O_7:Mn^{2+}$ elastomers generates intense yellow emission, easily visible to the naked eye (inserted in Fig. 5a). The collected ML spectra show emission from the $^4T_1-^6A_1$ transition of the $Mn^{2+}$ ions, consistent with the PL spectra of $Mn^{2+}$, while the blue emission from the $CaGa_4O_7$ host is not observed (Fig. 5a). The transient ML signals display an oscillatory output with a relatively stable intensity over approximately 30 cycles of mechanical friction, and the ML response demonstrates excellent reproducibility under cyclic photoexcitation charging (Fig. 5b), indicating a trap-controlled ML process[49]. Moreover, the ML intensity shows a linear increase with increasing mechanical pressure (Fig. 5c, d), exhibiting the characteristic of elastic ML. These findings suggest that the $CaGa_4O_7:Mn^{2+}$ elastomers can be effectively utilized to visualize and evaluate stress levels, providing an additional certified layer of stress-excited luminescence for advanced anti-counterfeiting measures.

Notably, the ML response exhibits highly durable output under long-term mechanical stimulus, with less than 40% attenuation of ML intensity observed after continuous 420 cycles of mechanical friction (Fig. 5e). The relative stability of the ML response in the short term is consistently observed, such as at the beginning and end of four hundreds of friction cycles (inserted in Fig. 5e). The durable ML behavior distinguishes the developed $CaGa_4O_7:Mn^{2+}$ phosphor from conventional trap-controlled ML materials, which typically experience a significant decay in ML intensity to an undetectable level within a few dozen cycles of mechanical stimuli[50]. These results support the suitability of $CaGa_4O_7:Mn^{2+}$ phosphor for high-level ML anticounterfeiting and encryption that require long-time delays and high mechanical durability.

## Studies of defect states

We began by examining the defect states associated with the blue emission in $CaGa_4O_7$, specifically considering unintentional impurity defects. This is because the intensity of the blue emission diminished when $CaGa_4O_7$ was prepared using $Ga_2O_3$ with higher purity (Supplementary Fig. 19). Inductively coupled plasma-mass spectrometer (ICP-MS) measurements of the prepared $CaGa_4O_7$ confirmed the presence of trace impurities (Supplementary Table 4). These results indicate that residual impurities present in the raw materials might contribute to the blue emission observed in the host material.

To further explore the luminescence performances of $CaGa_4O_7:Mn^{2+}$, we have carried out theoretical calculations to explore the contributions of intrinsic defects. For the host material $CaGa_4O_7$, the projected density of states (PDOSs) have been demonstrated, where the O-2$p$ orbitals dominate the valence band maximum (VBM) while the Ga-4$p$ orbitals are the main contributions of the conduction band minimum (CBM) (Fig. 6a). The 4$s$ orbitals of Ca sites mainly contribute to the valence band below the Fermi level. There is an evident gap of 4.60 eV between the VBM and CBM, which is further revealed by the bandstructure, supporting that the host is a broadband semiconductor as the experiment characterizations (Fig. 6b). With the introduction of the $Mn^{2+}$ dopants in the matrix, we notice that the overall PDOSs of the host materials have not been significantly affected (Fig. 6c). The $Mn^{2+}$ states are mainly located within the band gap. Moreover, we notice the Mn-3$d$ orbitals show an empty state above the VBM at $E_V$ + 2.21 eV ($E_V$ denotes 0 eV) and overlap with Ga-4$p$, which can induce an emission with a wavelength of 561 nm from this state to the occupied $Mn^{2+}$ state at $E_V$. The bandstructure of $CaGa_4O_7:Mn^{2+}$ reveals that although the VBM and CBM are downshifted, the bandgap size remains similar to the host material, which is supportive of the experiments (Fig. 6d).

Then, we systematically reveal the electronic structures of different intrinsic defects in the host materials. For the impurities, we notice that only 3$d$ orbitals of Cu impurity induce an empty state at $E_V$ + 2.61 eV, which is able to realize the blue emission with a wavelength of 475 nm (Supplementary Fig. 20a–d). In comparison, all the other impurities cannot improve the blue emissions since their states are deep in the VB and CB. These calculations were confirmed by the experimental results (Supplementary Fig. 21). For the intrinsic defects, we first investigated the interstitial (i) defects of Ca, Ga, and O (Supplementary Fig. 20e–g). Notably, the $i_{Ca}$ causes an occupied state at 2.78 eV above the VBM and $i_{Ga}$ influences the nearby O sites with an occupied state appearing near 1 eV above the VBM. In comparison, the neutral $i_O$ defect shows no evident change in the PDOS. Compared to interstitial defects, the antisite defects of cations display limited contributions to the luminescence performances (Supplementary Fig. 20h). Different positions of anion Frenkel (A-Fr) pair defects are also investigated, where two gap states are presented as deep electron traps (Supplementary Fig. 20i–l). Besides the interstitial defects, the cation vacancies (V) are another common type of defect in the oxides (Supplementary Fig. 20m–n). It is noted that $V_{Ca}$ and $V_{Ga}$ both induce unoccupied gap states at near 1.85 eV above the VBM. Beyond the simple cation vacancies, we also explore the vacancies of CaO and GaO, which represent the double vacancies of Ca/Ga with nearby O sites (Supplementary Fig. 20o–p). $V_{GaO}$ leads to the appearance of two empty gap states at 1.86 eV and 2.62 eV above the VBM, which is similar to the $V_{Ga}$. These empty gap states are able to enable the potential tunneling effect to promote the yellow emission with nearby $Mn^{2+}$

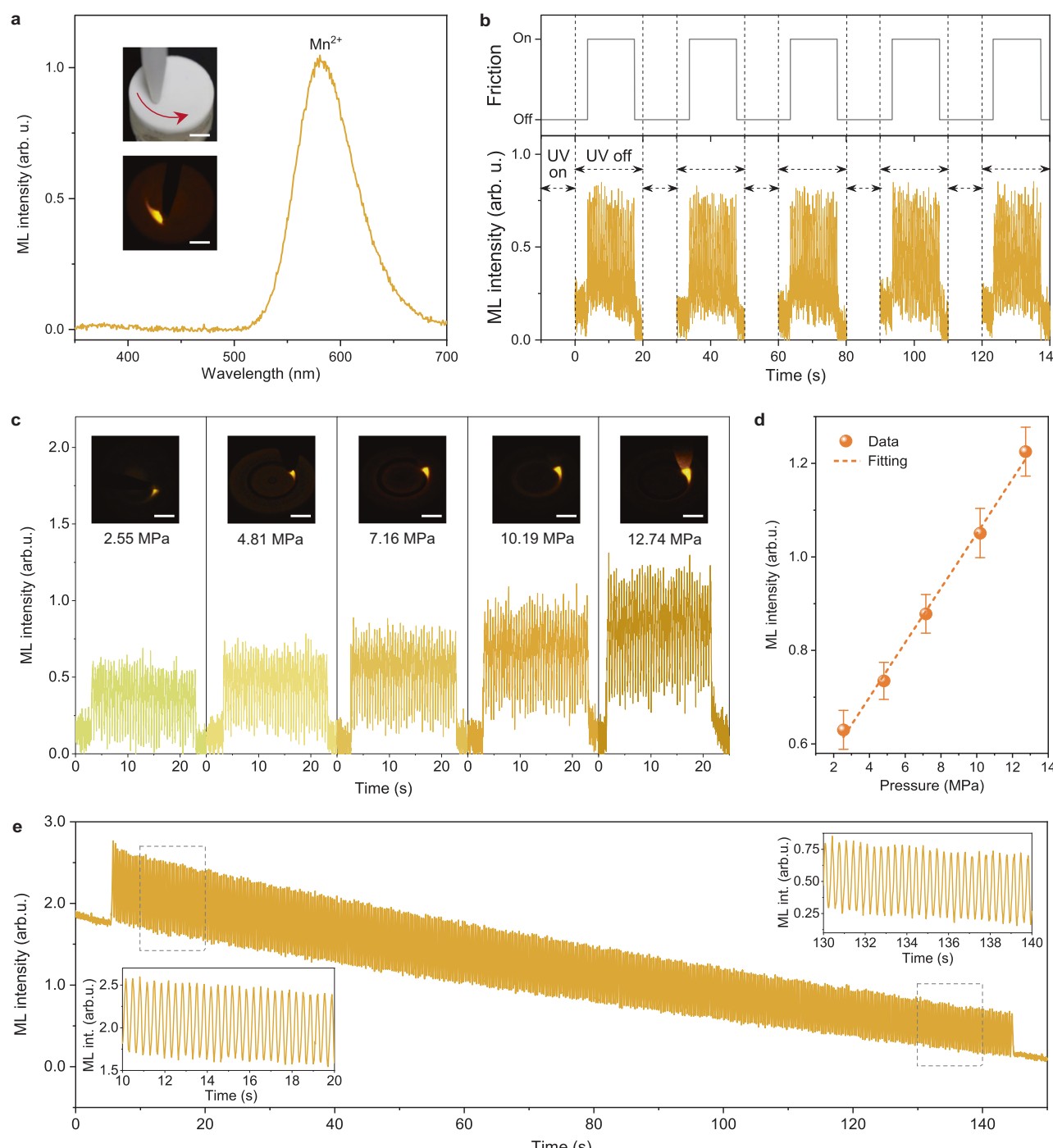

**Fig. 5 | ML properties of the phosphor/polymer composite elastomers. a** ML spectrum collected during the application of mechanical friction, with photographs of the friction setup (top) and the visually observed ML (bottom) as insets. The red curved arrow denotes the frictional direction of the rod. **b** Transient ML response to mechanical friction (7.16 MPa, 4π rad s⁻¹) after interval photoexcitation charging. Short dashed arrows between two adjacent vertical lines represent time periods when UV exposure is on, and long dashed arrows represent time periods

when UV exposure is off. **c** Transient ML response to mechanical friction with different pressures (2.55 – 12.74 MPa, 4π rad s⁻¹), with the ML photographs shown in insets. **d** Linear relation between ML intensity and pressure. Error bars represent standard deviation, $n = 10$ independent replicates. **e** Durable ML response under long-term cyclic mechanical friction (7.16 MPa, 6π rad s⁻¹). The insets show the enlarged ML response of the regions marked by boxes. Scale bars = 5 mm. Source data are provided as a Source Data file.

levels. Different from $V_{Ca}$, one occupied defect state at 0.26 eV above the VBM appears. As one of the most important defects in oxides, four different positions of $V_O$ with three different charge states (0, +1, +2) have been considered (Supplementary Fig. 22). $V_{O2}^{2+}$ and $V_{O4}$ have induced shallow empty gap states below the CBM. In contrast, other charged oxygen vacancies are promising contributors to the deep traps, which induce abundant deep unoccupied states in the range of

2.0–3.0 eV above VBM, acting as the intermediate states to promote electron transfer efficiency.

Based on the comprehensive screening of all the intrinsic defects, we have summarized all the gap states to identify the possible electron transfer pathway in the host materials (Fig. 6e). The intrinsic defects $i_O$, $V_{Ga}$, $V_{GaO}$, and $V_O$ dominate the electron transfer for the blue emission from the defect level to the ground state with photoemissions near

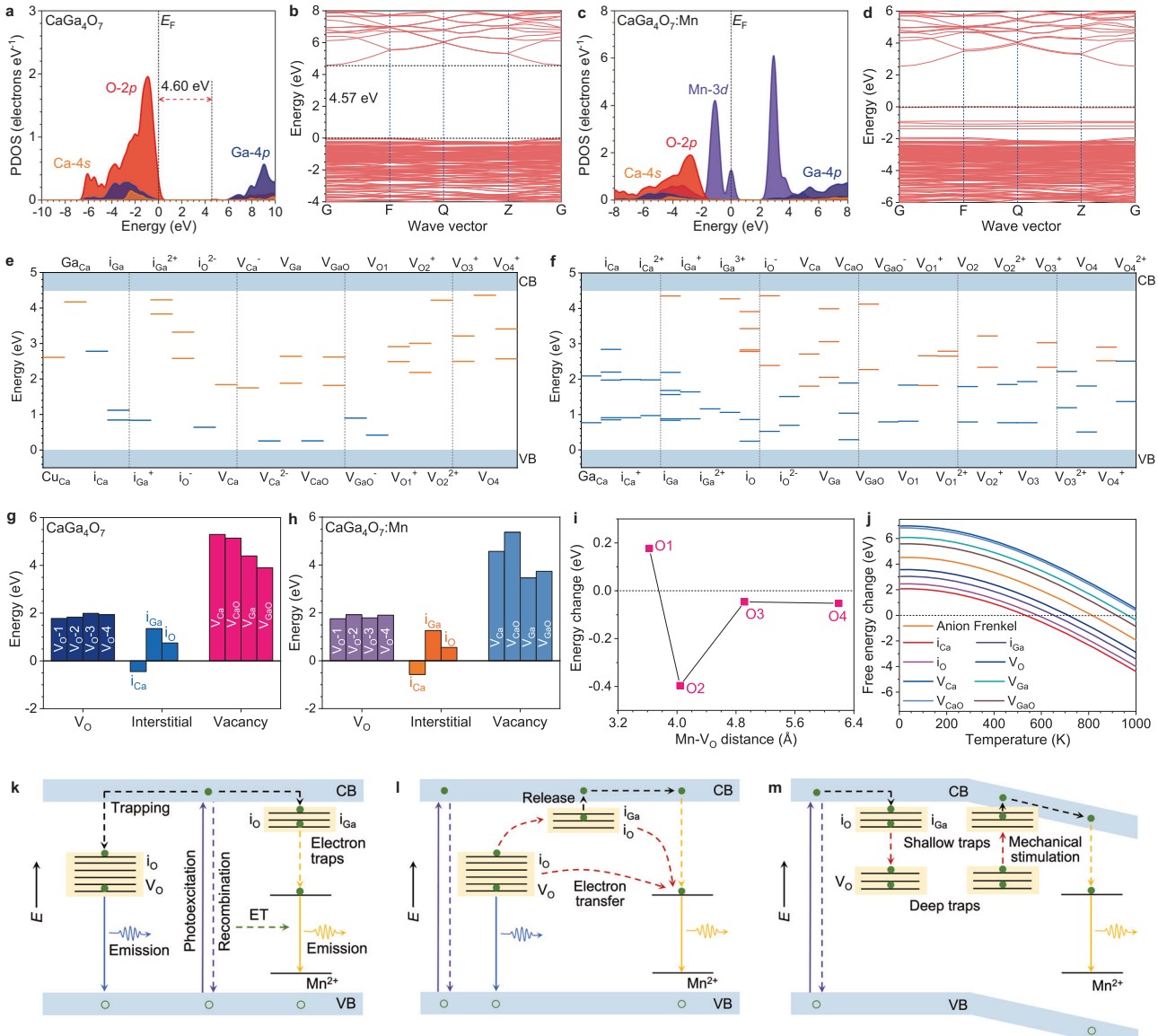

**Fig. 6 | Theoretical explorations of intrinsic defect assisted luminescence mechanism.** The (**a**) PDOS and (**b**) bandstructure of CaGa$_4$O$_7$. The (**c**) PDOS and (**d**) bandstructure of CaGa$_4$O$_7$:Mn$^{2+}$. The summarized defect levels in (**e**) CaGa$_4$O$_7$ and (**f**) CaGa$_4$O$_7$:Mn$^{2+}$. Orange lines = unoccupied states; blue lines = occupied states. The defect formation energies in (**g**) CaGa$_4$O$_7$ and (**h**) CaGa$_4$O$_7$:Mn$^{2+}$. **i** The formation energy changes of oxygen vacancies with different distances to Mn$^{2+}$. **j** The formation energy of defects with temperature changes. The mechanism diagram for (**k**) room-temperature PL, (**l**) high-temperature PL, and (**m**) ML. Green solid circles = electrons; green hollow circles = holes; purple solid arrows = photoexcitation; purple dashed arrows = recombination; green dashed arrows = energy transfer (ET); black dashed arrows = electron trapping or release; yellow dashed arrows = non-radiative transitions; yellow solid arrows = Mn$^{2+}$ emission; blue solid arrow = host emission; wine dashed arrows = electron transfer.

2.60 eV (476 nm). Meanwhile, the electron transfer from i$_{Ga}^{2+}$ to i$_{Ga}$ and V$_{O1}^{2+}$ to V$_{O1}$ also supplies blue emission with wavelengths of 457 nm and 490 nm, respectively. Moreover, the direct electron transfer from the impurity Cu$_{Ca}$ defect levels to the ground states also enables the photoemission around 2.61 eV (475 nm), which slightly enhances the blue emission due to the low concentration of impurity. For the CaGa$_4$O$_7$:Mn$^{2+}$, we have also summarized the main defect levels with positions close to the Mn$^{2+}$ based on the PDOSs screening results (Fig. 6f, Supplementary Fig. 23). It is noted that the introduction of Mn$^{2+}$ induces much higher defect levels, supplying abundant and flexible electron transfer pathways to achieve dynamic multicolor emission. There are abundant occupied states induced near the VBM, which reduces the electron transfer barriers to promote the overall luminescence performance. The i$_{Ga}$ and i$_O$ are the main contributions to the shallow traps below CBM and the oxygen vacancies mainly

induce deep defect levels near the mid-gap, where the empty states play as the traps for electron transfer and promote the tunneling effect.

To reveal the bandstructure evolutions under mechanical stimulations, we have considered two different situations by applying external strain and stress to CaGa$_4$O$_7$:Mn$^{2+}$ (Supplementary Fig. 24). Notably, with the external strain increasing from 0 to 5%, the bandstructure displays an evident tilt trend, while the energy levels of Mn$^{2+}$ remain at a similar position, which reduces the trap depth during the ML. Similarly, under different applied stresses, the energy levels of Mn$^{2+}$ are not affected. Meanwhile, to reveal the influences of Mn$^{2+}$ doping on the intrinsic defects, we have also explored the energetic trends (Fig. 6g–h). Compared to the pristine host materials, the introduction of Mn$^{2+}$ has reduced the formation energy costs of defects, which demonstrates that the defects are able to be stabilized

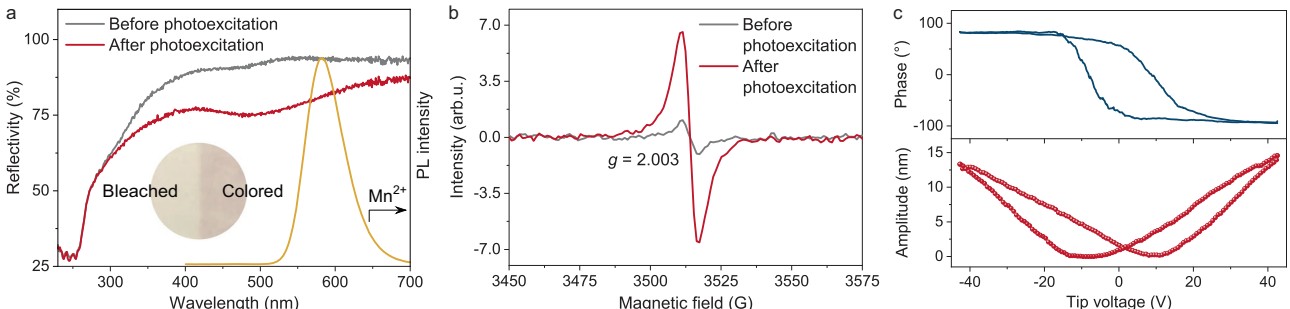

**Fig. 7 | Experimental investigations on color centers and local piezoelectricity.**
**a** Diffuse reflectance spectra of $CaGa_4O_7:Mn^{2+}$ before photoexcitation (gray line) and after photoexcitation (red line). The yellow curve shows the PL spectrum of $Mn^{2+}$ in $CaGa_4O_7:Mn^{2+}$. The inset shows an optical photograph of the $CaGa_4O_7:Mn^{2+}$ disc, with photoexcitation applied to the right half. **b** ESR spectra of $CaGa_4O_7:Mn^{2+}$ before photoexcitation (gray line) and after photoexcitation (red line). **c** PFM hysteresis loop of a $CaGa_4O_7:Mn^{2+}$ particle (top, phase signal; bottom, amplitude signal). Source data are provided as a Source Data file.

near the doping ions, especially the interstitial defects. In particular, $Mn^{2+}$ dopants lower the formation energy of most nearby oxygen vacancies with different distances (Fig. 6i). This confirms the synergistic contributions of doping and defect levels to the luminescence performances. To reveal the role of different defects, we have also investigated their formation energies regarding the temperature changes (Fig. 6j). Notably, interstitial defects are easily formed under temperature perturbations. As the temperature further increases, oxygen vacancies also become more, which modulates the mechanisms of temperature-dependent luminescence. In contrast, other types of defects display relatively high energy costs, which exhibit limited contributions to the luminescence.

Accordingly, we have demonstrated the luminescence mechanisms under different external stimulations in the single particle level summary (Fig. 6k–m)[51,52]. For the room temperature PL, the charge separation process is initiated by UV photoexcitation, where the electrons are excited from VB to CB, and the holes are left in the VB (Fig. 6k). For the $CaGa_4O_7:Mn^{2+}$, there are only limited defects formed during the synthesis, where $i_{Ga}$ and $i_O$ supply limited trap states near the CBM. The blue emissions of the host materials are mainly dominated by the defect states of $V_O$ near the mid-gap position at 2.5–2.8 eV above the VBM. The yellow emission of $Mn^{2+}$ is mainly induced by the energy transfer from the non-radiative recombination of electrons and holes. Since the defect states of $V_O$ are slightly higher than the state of $Mn^{2+}$, the electron transfer competition results in the suppression of yellow emission, realizing the dynamic color modulations. The high-temperature PL is strongly associated with the newly formed defects under thermal perturbations (Fig. 6l). As DFT calculations informed, the interstitial defects and oxygen vacancies will become much easier to form, especially in the range of 500–600 K. Under such a thermal perturbation, more abundant defect states are formed near VBM and the mid-gap, promoting the electron transfer from $V_O$ states to the shallow traps $i_{Ga}$ and $i_O$ states. This further strengthens the electron release from trap states to the $Mn^{2+}$ states, leading to enhanced yellow emissions with reduced blue emissions. For the ML, the local piezoelectric potential causes a tilt in the band structure, which is revealed by DFT calculations (Fig. 6m). Such a band tilt serves to reduce the trap depth, enabling the faster and more efficient electron release from the trap states to $Mn^{2+}$, thereby generating yellow luminescence. The presence of deep traps ensures a continuous supply of electrons to shallow traps during the ML. Therefore, cyclic mechanical stimulation applied to ML elastomers yields an intense ML with high mechanical durability.

## Physics and interactions in dynamic multimodal luminescence
In exploring the gradual weakening of the yellow emission upon photoexcitation, we initially considered the potential for photo-oxidation of $Mn^{2+}$ ions. The kinetics of the PL spectra show the emission intensity of $Mn^{2+}$ ions decays by more than 66% within 15 s

(Fig. 3h, j). If such a large attenuation was due to the photo-oxidation of $Mn^{2+}$ ions, a large percentage of high-valent Mn ions (e.g., $Mn^{3+}$, $Mn^{4+}$, or $Mn^{5+}$) would be formed. However, the investigation of PL spectra shows that only $Mn^{2+}$ emission was observed before and after photoexcitation, and no emission of other Mn ions was detected (Fig. 3b and Supplementary Fig. 2)[53–55]. XPS spectra show no change in the binding energy of $Mn2p_{3/2}$ or in the $Mn3s$ multiplet splitting value (5.4 eV) before and after photoexcitation (Supplementary Fig. 25). These experimental results confirm that photoexcitation did not modify the chemical state of the $Mn^{2+}$ ions[56,57].

After ruling out the possibility of photo-oxidation, the photochromic properties of $CaGa_4O_7:Mn^{2+}$ were investigated. We observed that photoexcitation (254 nm, 5 s–3 min) led to a distinct and rapid change in the apparent color from milky white to brown, as confirmed by a significant decrease in reflectance in the visible region (Fig. 7a). Upon cessation of the photoexcitation, a gradual decolorization process occurred, with the reflectance gradually recovering (Supplementary Fig. 26). These optical observations indicate the formation of a color center during the photoexcitation process. The yellow emission of $Mn^{2+}$ (500–700 nm) falls within the absorption range (300–700 nm) of the color center, suggesting an energy transfer from $Mn^{2+}$ to the color center[58,59]. The energy transfer results in a reduction in the intensity of the yellow emission. Additionally, the analysis of electron spin resonance (ESR) spectra reveals a significant increase in the characteristic signal of the single ionized oxygen vacancies at $g = 2.003$ upon photoexcitation (Fig. 7b and Supplementary Fig. 27). This finding indicates that oxygen vacancies within the material trap the photoexcited electrons, leading to the formation of the color center[60]. These experimental results align with the theoretical calculations emphasizing the role of oxygen vacancies as electron traps. Significantly, these findings offer an explanation for the observed correlation between the emission color-changing rate and the re-excitation interval as well as the optical excitation power. Firstly, in response to the perturbation of ambient temperature, electrons in the traps are continuously released (Supplementary Fig. 26). This phenomenon introduces variations in the population of electrons within the traps over the re-excitation interval, influencing the duration required for trap refilling and, consequently, impacting the emission-color changing rate. Secondly, at higher photoexcitation powers, trap filling is accelerated, resulting in a faster emission color-changing rate.

To probe the driving mechanism behind ML, ML tests were conducted using five materials with different electron affinities as friction bars. The results show that the ML intensity is essentially uncorrelated with the electron affinity of the friction materials (Supplementary Fig. 28), ruling out the possibility of triboelectrically driven ML. Subsequently, piezoresponse force microscope (PFM) measurements were performed on random individual particles of $CaGa_4O_7:Mn^{2+}$. These measurements reveal typical butterfly-like curves and hysteresis

loops with a phase contrast of 180° (Fig. 7c and Supplementary Fig. 29). These characteristics indicate a polarization reversal, providing evidence for the presence of local piezoelectricity and unveiling the origin of the piezoelectrically driven ML[61]. Considering the centrosymmetric structure of $CaGa_4O_7$:$Mn^{2+}$ (space group $C2/c$), the observed piezoelectricity can be attributed to local symmetry breaking caused by structural defects[62,63].

## Applications in dynamic color/pattern display

The dynamic multicolor evolution and photo-thermo-mechanical responsiveness of the developed $CaGa_4O_7$:$Mn^{2+}$ make it an appealing candidate for applications in multidimensional anti-counterfeiting and multimodal sensing (Fig. 8). To demonstrate its potential, we fabricated solar-shaped composite films using a simple mold-forming method (Fig. 8a and Supplementary Fig. 30). Silicone rubber was chosen as the elastomeric matrix due to its exceptional combination of thermal-mechanical-optical properties, including thermal stability (up to 573 K, Supplementary Fig. 31), mechanical stretchability (over 300%, Supplementary Fig. 32) and optical transparency (over 88% in the 400−700 nm range, Supplementary Fig. 33). The circular body of the solar-shaped film consisted of $CaGa_4O_7$:$Mn^{2+}$ particles, while the surrounding pattern featured red-emitting $NaNbO_3$:$Pr^{3+}$ particles. This arrangement highlighted a contrast between the dynamic multicolor evolution of the former and both the consistent emission color, as well as the thermally quenched PL of the latter (Supplementary Fig. 34). These composite films offer flexibility, water-resistance, and high-temperature resistance (up to 573 K) (Fig. 8b), making them suitable for use in diverse environments.

For a proof-of-principle demonstration of the solar-shaped film's capabilities, we initiated the process by exciting it using a handheld 254 nm lamp at room temperature (Fig. 8c). Within just a few seconds, the emission color of the circular body changed from yellow to cool white, as depicted by the variation of the Commission internationale de l'éclairage (CIE) chromaticity coordinates with photoexcitation time (Fig. 8f). In contrast, the red-emitting pattern remained unchanged in color or intensity. These dynamic PL images appear to mimic the color change of a rising sun and can serve as a temporal feature for anti-counterfeiting (Supplementary Movie 1). Furthermore, while keeping the film under photoexcitation, we applied thermal stimulus using a heating table (Fig. 8d). As the temperature gradually increased from 298 K to 348 K, the emission color of the circular body reversed from cool white to yellow, as indicated by the temperature dependence of the CIE chromaticity coordinates (Fig. 8f). With a further temperature increase to 473 K, the brightness of the yellow emission became more pronounced. Even when elevated to 573 K, the naked eye could still perceive the exceptionally bright yellow emission. The emission ratios of yellow emission to blue emission ($I_{579}/I_{474}$) followed an exponential dependence on temperature and could be utilized for quantitatively sensing the ambient temperature (Fig. 8g and Supplementary Fig. 35). In contrast, during the temperature rises from 298 K to 573 K, the red emission gradually diminished and eventually disappeared entirely. These PL images captured during heating depict sunset-like color and pattern changes (Supplementary Movie 2) that can provide temperature-related spatial features for anti-counterfeiting measures. Additionally, we examined the film's ability to visualize dynamic stress distribution by writing on it using a ballpoint pen at room temperature (Fig. 8e). The ML profile clearly reflected the trajectory of the pen tip, enabling the observation of distinct handwriting graphics (Supplementary Movie 3), such as the ML pattern of "LUMINESCENCE" produced while writing letters. By leveraging the quantitative relationship between ML intensity and stress intensity, the relative ML brightness emerges as a valuable indicator of stress information at diverse locations (Fig. 8h). This characteristic encompasses a stress-related spatial feature, adding an additional layer of complexity that renders counterfeiting more challenging.

Our user-friendly mold-forming method facilitates the fabrication of composites in diverse shapes (Supplementary Fig. 36). Furthermore, by utilizing different combinations of phosphors, we can showcase individually designed luminescence responses to multiple stimuli, as well as dynamic color and pattern evolution (Supplementary Fig. 37). These experiments exemplify the flexibility and versatility of our proposed strategy for multi-dimensional displays, multilevel anti-counterfeiting measures and multimodal sensing applications.

## Discussion

Our study introduces a design strategy for developing dynamic multimodal phosphors by incorporating transition metal ions into self-activated gallate materials. The resulting $CaGa_4O_7$:$Mn^{2+}$ phosphor showcases time-dependent dynamic multicolor evolution, and also modulation in emission color-changing rates through re-excitation intervals and photoexcitation powers. Additionally, the phosphor displays an anti-thermal-quenched $Mn^{2+}$ emission during temperature-dependent static multicolor reversal, along with a reproducible elastic ML characterized by high mechanical durability. Extensive insights from DFT calculations have explored comprehensive defect levels and their interactions with $Mn^{2+}$ dopants. The defect levels not only dominate the blue emissions of the host materials but also offer abundant trap states to facilitate the electron transfer with $Mn^{2+}$ levels, thus enabling the dynamic multicolor luminescence and multimodal emission under different excitations. Furthermore, we have demonstrated the practical viability of this phosphor by developing patterned composite films that possess flexibility, water immersion stability, and high-temperature resistance. The dynamic multimodal luminescence of the developed phosphor exhibits evolving colors and patterns under photo-thermal-mechanical stimuli in both temporal and spatial dimensions. Our findings establish a foundation for the systematic design of multi-responsive, dynamic color-changing phosphors by strategically combining self-activated hosts and luminescent dopants. This research is expected to have significant implications on multi-dimensional dynamic photonic displays, advanced multilevel anti-counterfeiting/encryption techniques, and multi-stimulus sensing applications.

## Methods

### Phosphors

The phosphors with the chemical formula $Ca_{1-x}Mn_xGa_4O_7$ ($Ca_{1-x}Ga_4O_7$:$xMn^{2+}$, $x = 0$, $0.2 \times 10^{-4}$, $1 \times 10^{-4}$, $2 \times 10^{-4}$, $3 \times 10^{-4}$, $4 \times 10^{-4}$, $6 \times 10^{-4}$, and $10 \times 10^{-4}$) were synthesized through solid-state reactions. Stoichiometric mixtures of high-purity $CaCO_3$ (99.99%), $Ga_2O_3$ (99.99%), and $MnCl_2$ (99.99%) were thoroughly ground and compacted into discs measuring 10 mm in diameter and 1 mm in thickness under a pressure of 10 MPa. These discs were sintered at 1573 K for 4 h in an argon atmosphere using a tube furnace (BTF-1600C, Anhui BEQ). Following the cooling process to room temperature, the phosphor discs underwent pulverization, grinding, and sieving through a 20 μm sieve to yield microparticles suitable for subsequent use. $NaNbO_3$:$Pr^{3+}$ phosphor was synthesized through $Na_2CO_3$ (99.99%), $Nb_2O_5$ (99.99%) and $Pr_6O_{11}$ (99.99%) by a solid-state reaction (1573 K, 3 h in air). A moderate amount of Na was substituted by 0.3 mol% Pr.

### Composite materials

To examine the ML properties of the phosphors, cylindrical composites consisting of phosphor and epoxy resin were prepared, with a diameter of 25 mm and a thickness of 15 mm. The phosphor microparticles were embedded into an optical epoxy resin (SpeciFix, Struers GmbH) at a weight ratio of 1:9, followed by curing at 333 K for 2 h. To select a suitable organic substrate for demonstrating the multifunctional applications of the developed phosphors, five heat-resistant organic polymers were used to prepare the

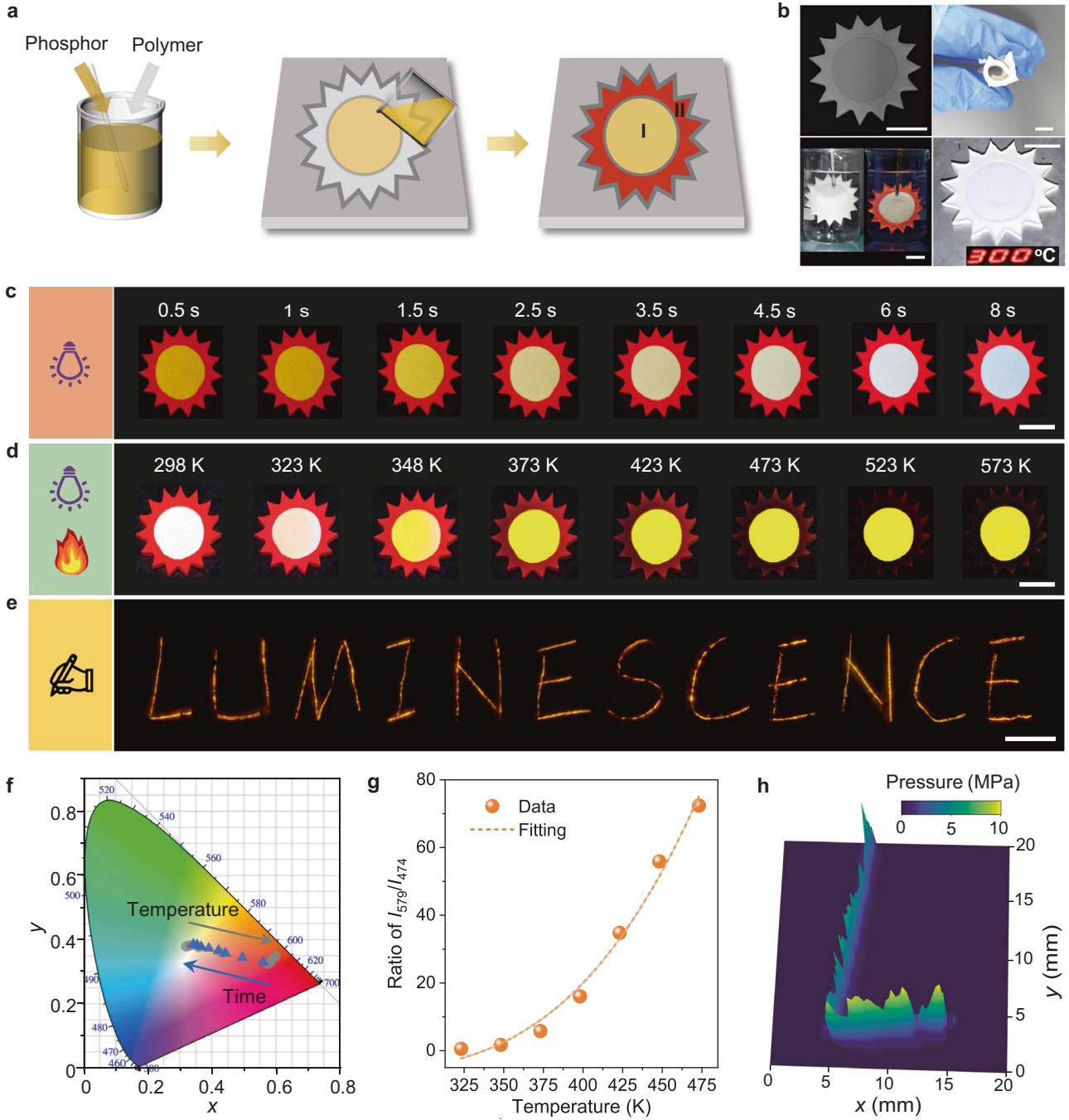

**Fig. 8 | Multi-responsive multicolor luminescence for multidimensional dynamic displays and multimodal sensing. a** Illustration of the preparation process for solar-shaped composite films. Two phosphors, $CaGa_4O_7:Mn^{2+}$ (I) and $NaNbO_3:Pr^{3+}$ (II), are embedded in Region I and Region II, respectively. **b** Photographs of the prepared composite film (top left), showing its flexibility (top right), water-immersion stability (bottom left: bright-field image and dark-field PL), and thermal stability up to 573 K (bottom right). **c** Time-dependent multicolor PL patterns excited by a handheld 254 nm lamp. **d** Temperature-dependent multicolor PL patterns and thermally boosted emission. **e** "LUMINESCENCE" ML patterns triggered by handwriting letters in Region I of the composite film. **f** Dependence of the CIE chromaticity coordinates on photoexcitation time (0.5–8 s, indicated by blue triangles) and temperature (298–573 K, indicated by gray circles). **g** Dependence of the emission intensity ratios of $I_{579}/I_{474}$ on temperature. Source data are provided as a Source Data file. **h** Pressure mapping of the handwritten "L" trajectory. Scale bars = 10 mm.

$CaGa_4O_7:Mn^{2+}$/polymer composite films, including silicone rubber (Xinbang Chemical Technology), PDMS (polydimethylsiloxane, Dow Corning Shanghai), RTVS605 (dimethylvinylated and trimethylated silica + hydrogen-terminated vinyl silicone polymer + vinyl silicone polymer + polydimethylsiloxane, Hasuncast Industries), HL1028 (polydimethylsiloxane + vinyl silicone oil, Shenzhen Huasheng Tongchuang Technology), and TY866 (silica powder + thermal conductive powder + polydimethylsiloxane, Dongguan Tianyu Composite Materials). The phosphor microparticles were incorporated into the organic polymers at a weight ratio of 2:1. The curing conditions for these composite films were as follows: silicone rubber-based film cured at 333 K for 10 min, PDMS-based films cured at 353 K for 2 h, RTVS605-based films cured at 373 K for 30 min, HL1028-based films cured at 353 K for 15 min, and TY866-based films cured at 353 K for 360 min.

## Structural characterization

Room-temperature and variable temperature XRD patterns were acquired using a Rigaku SmartLab X-ray diffractometer (9 KW) by CuKα radiation ($\lambda = 1.5406$ Å). The crystal structures were refined employing Rietveld refinement method through the General Structure Analysis System program. Scanning electron microscope (SEM) images were captured using a JEOL JSM-7800F microscope. HRTEM observations, SAED patterns, HAADF-STEM images, elemental mapping distribution, and energy dispersive X-ray spectrometry (EDS) were obtained utilizing a JEOL JEMF200 microscope. For the preparation of TEM samples, the phosphors were dispersed in ethanol, then dropped on a copper grid and dried on a hot plate (423 K), and tested on small crystals. Impurity element analysis was performed using an ICP-MS spectrometer equipped with an Agilent 7850 spectrometer. X-ray photoelectron spectroscopy (XPS) analysis was performed on a Thermo Fischer ESCALAB 250Xi XPS microprobe with monochromatic Al Kα (h$\nu$ = 1486.6 eV) radiation as the X-ray source. Electron paramagnetic resonance (EPR) spectra were measured using a Bruker A300 spectrometer operating at a frequency of 9.2 GHz. PFM hysteresis loops were measured using a Bruker Multimode 8 atomic force microscope.

## Optical characterization

Steady-state PL spectra were obtained using a fluorescence spectrometer (F-4700, Hitachi) equipped with a 150 W Xe lamp and a heating-cooling stage (CH600-190, GoGo Instruments). Note that steady-state PL spectra were acquired on the phosphors that have reached stable emission. Time-resolved PL spectra were recorded using a fiber spectrometer with a silicon-based CCD detector (QEPro, Ocean Optics), and the gate time for spectral acquisition was set at 350 ms. The excitation light for time-resolved PL spectra was supplied by a fluorescence spectrometer (F-4700, Hitachi) with a 150 W Xe lamp. A portable hand-held UV lamp (8 W, 254 nm) was employed to irradiate the samples while capturing PL photographs. Additionally, it was used to charge the traps of the samples and induce photochromism. The power density of photoexcitation was assessed using an optical power meter (PM100D, Thorlabs) and adjusted within the range of $0.25 - 0.75$ W m$^{-2}$ by varying the distance between the lamps and samples. Unless stated otherwise, the power density of photoexcitation was maintained at 0.50 W m$^{-2}$. The PL lifetimes and PLQY values were determined using a spectrofluorometer (FLS1000, Edinburgh) equipped with both a 450 W Xe lamp and a μFlash lamp. For PL lifetime measurements, a total of 10,000 photons were collected. Note that the phosphors underwent heat treatment to evacuate traps before PL measurements. Diffuse reflectance spectra, transmission spectra, and absorption spectra were recorded using a UV-Vis spectrophotometer (TU-1901, Persee). ThL curves were measured using a ThL meter (FJ427A1, Beijing Nuclear Instrument Factory). For ML measurements, mechanical stimulation was applied using a friction testing machine (MS-T3001, Lanzhou Huahui), and the transient ML signals were collected using a photon counting system consisting of a computer-controlled photon counting probe (H10682-01, Hamamatsu Photonics) and a photon counting module. Five materials were used for the friction rods, including polyamide (PA), polypropylene (PP), polyformaldehyde (POM), polyethylene terephthalate (PET), and polyvinyl chloride (PVC). If not specified, a friction rod made of PP was used. ThL spectra and ML spectra were recorded using a fiber spectrometer (QEPro, Ocean Optics). Digital photographs were captured using a Sony α7R II camera equipped with an FE 2.8/90 lens. All measurements, excluding heating-related experiments, were conducted at standard room temperature.

## Calculation setup

We have applied the DFT + U method by CASTEP code[64]. To improve the accuracy of the electronic structure calculations, the Hubbard U parameters have been introduced in the Perdew-Burke-Ernzerhof (PBE) + U calculations with a kinetic cutoff energy of 880 eV based on the Anisimov-type rotational invariant DFT + U method[65]. To minimize the effect of the localized hole states produced by the 2p orbitals of the O sites, the self-consistently determined Hubbard U potentials have also been applied to the O-2p orbitals as many previous works of oxide materials[66–69]. We chose the (3s, 3p, 3d, 4s) states as the valence states of Ca, (3d, 4s, 4p) for Mn, (4s, 4p) for Ga, and (2s, 2p) for O. The norm-conserving pseudopotential is accordingly generated based on the OPIUM code in the Kleinman-Bylander form to reduce the systematic error induced by the overlapping of the atomic core-valence electron densities[70]. Meanwhile, the RRKJ method has been used for the optimization of pseudopotentials[71]. To guarantee electronic minimization and convergence, we introduce the ensemble DFT (EDFT) method to solve the Kohn-Sham equation[72]. The reciprocal space integration was performed by k-point sampling with a grid of $10 \times 10 \times 10$ k points for the CaGa$_4$O$_7$ unit cell, and a grid of $3 \times 3 \times 5$ for defect electronic structure calculations in the supercells. The convergence criteria have been set as follows: the total energy should be smaller than $5.0 \times 10^{-7}$ eV per atom; the Hellmann–Feynman force should be lower than 0.01 eV Å$^{-1}$ per atom. With our self-consistent determination process[73–80], the on-site Hubbard U parameters are 2.513, 5.013 and 2.700 eV for Ca, Ga and Mn, respectively.

## Reporting summary

Further information on research design is available in the Nature Portfolio Reporting Summary linked to this article.

## Data availability

The data that support the findings of this study are available from the corresponding authors upon request. Source data are provided with this paper.

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

## Acknowledgements

J.-C.Z. acknowledges support from the National Natural Science Foundations of China (grant nos. 11774189 and 12374387), the Taishan Scholar Program (grant no. tsqn202211057), the Natural Science Foundation of Shandong Province (grant no. ZR2023MA075), and Fundamental Research Funds for the Central Universities (grant nos. 209202141002 and 202364006). M.W. and J.-C.Z acknowledge support from the National Natural Science Foundations of China (grant no. U22A20135). B.H. acknowledges support from the National Natural Science Foundation of China/Research Grant Council of Hong Kong Joint Research Scheme (grant no. N_PolyU502/21), National Natural Science Foundation of China/Research Grants Council of Hong Kong Collaborative Research Scheme (grant no. CRS_PolyU504/22), the funding for Projects of Strategic Importance of The Hong Kong Polytechnic University (grant no. 1-ZE2V), Shenzhen Fundamental Research Scheme-General Program (grant no. JCYJ20220531090807017), the Natural Science Foundation of Guangdong Province (grant no. 2023A1515012219) and Departmental General Research Fund of The Hong Kong Polytechnic University (grant no. ZVUL). S.W. acknowledges support from the National Natural Science Foundations of China (grant no. 62375248). B.H. thanks the support from Research Center for Carbon-Strategic Catalysis (RC-CSC), Research Institute for Smart Energy (RISE), and Research Institute for Intelligent Wearable Systems (RI-IWEAR) of the Hong Kong Polytechnic University.

## Author contributions

J.-C.Z. conceived and supervised the research, and wrote the manuscript. Y.T. and Y.C. performed the main experiments. B.H., M.S. and K.D. performed the DFT calculations and analysis. J.C., W.L. and S.W. contributed to a portion of the experiments and analysis. X.L., J.Q., L.Z. and M.W. participated in the discussion of results and revision of the manuscript. All authors contributed to the final manuscript.

## Competing interests

The authors declare no competing interests.

## Additional information

[1]College of Physics and Optoelectronic Engineering, Faculty of Information Science and Engineering, Ocean University of China, Qingdao 266100, China. [2]Engineering Research Center of Advanced Marine Physical Instruments and Equipment of Education Ministry of China, and Key Laboratory of Optics and Optoelectronics of Qingdao, Ocean University of China, Qingdao 266100, China. [3]Department of Applied Biology and Chemical Technology, The Hong Kong Polytechnic University, Hong Kong SAR, China. [4]College of Optical Science and Engineering, State Key Laboratory of Modern Optical Instrumentation, Zhejiang University, Hangzhou 310027, China. [5]School of Chemical Engineering and Technology, Sun Yat-sen University, Zhuhai 519082, China. [6]These authors contributed equally: Yiqian Tang, Yiyu Cai. ✉e-mail: bhuang@polyu.edu.hk; zhangjuncheng@ouc.edu.cn

