## [Peer Review File · Nature Communications]

Dynamic multicolor emissions of multimodal phosphors by Mn²⁺ trace doping in self-activated CaGa₄O₇Reviewers' Comments:

Reviewer #1:

Remarks to the Author:

The manuscript entitled "Enabling Dynamic Multicolor Emission and Photo-Thermo-Mechanically Responsive Luminescence via Trace Mn^{2+} Doping in Self-Activated $CaGa_4O_7$." written by Y. Tang *et al.* reports on the unconventional optical properties of $CaGa_4O_7$ doped with Mn^{2+} . Phosphors were elaborated following a high temperature solid-state reaction, leading to micrometric powder. Prepared phosphors demonstrate intense luminescence with interesting unconventional properties: dynamic color change, thermal color change, thermally enhanced luminescence and mechanoluminescence. An almost complete set of optical characterization is proposed to describe these properties and theoretical calculations are coupled to additional characterization methods to unravel the origin of the observed optical properties. Finally, authors embedded the developed phosphor in a polymer matrix to demonstrate interesting anticounterfeiting perspectives and place their material into an applicative context. I personally think that the proposed work might interest readers from material science and phosphor communities. I therefore recommend this work to be published in Nature Communication but I would like to have the following queries resolved before.

1. In the beginning of the articles, authors tune Mn^{2+} concentration to vary phosphor color. As, latter on, it is demonstrated that the color varies with source characteristics (excitation time for instance), I would suggest to specify excitation characteristics in figure caption (Figure S2 here). This might be done systematically.
2. Do authors have an idea of the typical luminescence efficiency of developed phosphors? For instance, when doping content is varied I regret that PLQY is not discussed.
3. Various spectral evolution of $CaGa_4O_7$ and $CaGa_4O_7:Mn^{2+}$ emission with time and temperature are shown all along the manuscript (Fig.3, Fig.4). I think that CIE diagrams can be placed here to ease readability.
4. Authors discuss on the evolution of blue and yellow emissions kinetics with excitation power. I would like to see the profiles of this kinetics as it is done for the experiment with re-excitations.
5. Authors observed an increase in Mn^{2+} emission with temperature. Can authors comment on the evolution of the luminescence efficiency with temperature?
6. A spectral shift of Mn^{2+} band is observed with temperature, associated with a thermally induced lattice deformation. Still a plateau can be observed from 300 to 400 K. How can authors explain this plateau? Please add some references.
7. To discuss on the origin of the blue luminescence from host material, authors investigated potential impurities and from simulations concluded that blue emission might originate from electron-hole recombination from impurity level (Si_{Ga}^+ , Al_{Ga} , Zn_{Ca} or Ge_{Ga}^+) to V_{Ga}^- defect level. Is it possible to prove these conclusions? Can authors imagine intentionally doping $CaGa_4O_7$ matrices by Si, Al, Zn and Ge to reveal the genuine origin of the blue emission? If the real origin of the blue emission is found, can it be used to enhance it and tune the dynamic luminescence properties?
8. I think that PL decays of the blue and yellow emission can bring interesting information when discussing the luminescence mechanisms.

9. Why can the traps recombine only on Mn^{2+} centers and not on defects responsible of blue emission? Is it due to a local charge recombination? Can simulations prove that oxygen vacancies are more stable in the vicinity of Mn^{2+} impurity? Did authors record ThL of undoped samples? If yes, electrons might not go through the CB as represented in the scheme 6J.

10. Can authors develop on the explanation of the decrease of the yellow (Mn^{2+}) emission with excitation time? I personally find this part a little confusing. It is said that filling of traps (Vo^{2+} , Vo^{+}) compete with yellow emission, leading to the decrease of yellow emission. When traps are fully filled one can expect that the competition between the two process will no longer occur, leading into more intense yellow emission, a typical for persistent phosphors. How can observed results be placed in the context of persistent phosphors loading curves (10.1002/adpr.202100179, 10.1364/OME.6.000844)? Is it possible that available Mn^{2+} ions diminish through photo-oxidation due to electron trapping?

11. A tilt in the band structure is observed under mechanical stimulation. Can Mn^{2+} emission band (4T_1) energy be shifted through mechanical stimulation too?

12. In the final "applications" part, the developed phosphor is embedded in silicone rubber, chosen for its thermal, mechanical as well as optical transparency characteristics. However, the proposed flexible matrix demonstrates strong absorption in the UV range below 300 nm -see Fig. S22-. Is it the best matrix knowing a 254 nm UV lamp is used to excite the phosphor? The optical power density felt by the phosphor will therefore be different compared to the powdered phosphor. Do authors think that this is a potential application issue knowing that the dynamic optical properties of the developed phosphor depends on the source intensity -see Fig. S6-? Also, authors should be careful when mentioning the potentiality of optical sensing of ambient temperature. How is the silicone rubber matrix influencing the thermal conductivity of the sensor and the emission ratio (I_{579}/I_{474})? Indeed, its absorption at 579 nm and 474 nm is different. I would therefore like to see similar "Temperature-dependent luminescent intensity ratio and relative sensitivity" conducted on $CaGa_4O_7:Mn^{2+}$ devoid of polymer matrix.

Reviewer #2:

Remarks to the Author:

The manuscript "Enabling Dynamic Multicolor Emission and Photo-Thermo-Mechanically Responsive Luminescence via Trace Mn^{2+} Doping in Self-Activated $CaGa_4O_7$ " discusses the optical properties of Mn-doped gallate crystals. The authors show that these crystals exhibit unique time-dependent dual luminescence, temperature-dependent emission, and mechanoluminescence, which could have valuable applications in the detector and anti-counterfeiting industry. The experimental data presented in the manuscript supports the authors' claims. However, the data quality and analysis require significant improvements before definite recommendation for publication.

The major issues are listed below:

The authors should include an analysis of the presence of Mn⁴ in their experimental results, as different oxidation states of Mn have different optical properties. Additionally, the statement "Photoluminescent excitation (PLE) spectra, monitored at different emission wavelengths ranging from 400 to 625 nm, exhibit identical shapes and features, with an excitation peak observed at 256 nm (Fig. 3c). This result indicates that the blue and yellow emissions arise from the relaxation of the same excited state" lacks precision. While the VB-CB transition might be dominant under the above bandgap excitation, a PLE spectrum extended to the visible range should show the atomic level transitions of the defects and the Mn dopant. The PLE peaks related to such transitions might be significantly weaker than the direct band gap transition of the host due to the forbidden transitions between the electron

levels of the Mn ions.

The characterization of the material's structure must include high-quality data. Figure 2 displays a high-resolution transmission electron microscopy (HR-TEM) image of the crystal edge and selected area electron diffraction (SEAD), but the data is challenging to comprehend. There is no information on how the sample preparation for HR-TEM was carried out, whether it was a focused ion beam (FIB) slice, or a small crystal found under the TEM.

In regard to high-angle annular dark-field scanning transmission electron microscopy (HAADF-STEM), the elemental distribution is not uniform and lacks explanation. The subfigure labeled "C" next to the STEM image is slightly confusing next to the elemental mapping. The X-ray diffraction (XRD) in the supplemental information (SI) shows a smaller range, and considering that gallates exist in different compositions, the XRD and quality factor of the Rietveld fitting should be reported.

Although the experimental data is relatively well documented, the explanation of the physical process seems more speculative than experimentally supported. The $4T_1 \rightarrow 6A_1$ transition is parity and spin forbidden, meaning that any other relaxation path is more likely. The UV excitation enables the excited electrons from the Mn ground state to be delocalized in the conduction band (CB) and locate a non-forbidden relaxation process. This process could change the Mn ion's oxidation (or charge) state, reducing the photoluminescence (PL) intensity. The electron paramagnetic resonance (EPR) and photo-EPR data reveal the generation of EPR active defects upon illumination, which may be oxygen vacancies in some of their charge states. Both the oxygen vacancy and the change of the oxidation state of Mn dopants change the color of metallic oxides. We strongly encourage the authors to review the description of the physical process written in the manuscript to explain the temporal and other changes in the optical properties. A more detailed EPR and photoluminescence excitation (PLE) study can answer most questions. If oxygen vacancies are generated upon UV excitation, those vacancies are likely to be permanent, and the EPR signal of the vacancy should evolve and show the same intensity in the dark. The change of the Mn oxidation state can also be followed by EPR and PLE. However, if the signal increase in photo-EPR is due to a charging effect, the signal should decrease rapidly in the dark.

Density functional theory (DFT) calculations identified defect levels close to the conduction band minimum (CBM) for Si, Al, Zn, and Ge substituting Ga defects. The authors also found a deep occupied level for the Ga-vacancy? It is unclear what V_{GaO} means? A double vacancy or single vacancy of gallium? The authors compared the energy difference between these Kohn-Sham levels and the experimentally observed maximum intensity in the PL spectrum (~ 2.6 eV). For the yellow luminescence, the authors suggest that the Mn substituting Ga defect interacts with the O-vacancy where the former creates defect levels close to CBM whereas the different charge states of O-vacancy create levels close to CBM and deep in the band gap. Unfortunately, the authors do not define whether these defect levels are empty or occupied which makes it difficult to understand the arguments of the authors. Again, the Kohn-Sham level of the positively charged O-vacancy is directly compared to the observed trap level at ~ 2.3 eV.

As will be explained below, the comparison of DFT results to experiments stands on very weak feet, so it is questionable whether these DFT results can be applied for the discussion of the excitation and de-excitation processes.

In Figs. 6l-n, the authors mix the single particle picture (band diagram) and the many-body levels (e.g., $6A_1$, $4T_1$). This is wrong as explained in Refs. [Phys. Rev. B, 104, 235301 (2021), Nanophotonics 12, 359 (2023)].

The authors did not define which functional did they apply. This is a red flag against publication to any scientific journal. The authors mention that they applied DFT-1/2 method for self-interaction correction. The validity of this method is restricted. Many-body perturbation methods and wave

functions methods are robust methods to heal the potential band gap underestimation of the host material by the applied DFT functional and the failure in the accurate position of the defects' ionization energies. The authors may calculate the charge transition levels in a traditional fashion with applying charge correction for the charged defect species with a well-chosen functional and/or posteriori correction methods.

In the excitation process, the electron-hole interaction will determine the final excitation energy (e.g., can be calculated by Bethe-Salpeter equation) and the energy difference between Kohn-Sham levels is a rather poor estimation for the excitation energy. Furthermore, if two defects are involved in the excitation then the electron-hole interaction energy depends on the distance between the donor and acceptor defects. The authors did not discuss this issue at all. In addition, the maximum of the PL signal may be associated with the phonon participation in the luminescence process which is again completely ignored by the authors (this effect can be very large in these materials). The overall impression about the application of DFT theory is that the authors do not understand deep enough the defect physics at all, and they oversimplify the picture to find some supportive data for their experimental findings. In fact, the quality of the DFT studies and the validation of the results are very far from the point where a direct comparison to the experimental results can be reliably done.

Minor issues:

Line 146: 'x' is missing in the composition

Reviewer #3:

Remarks to the Author:

I co-reviewed this manuscript with one of the reviewers who provided the listed reports. This is part of the Nature Communications initiative to facilitate training in peer review and to provide appropriate recognition for Early Career Researchers who co-review manuscripts

Response to Reviewers

Dear Reviewers,

Thank you for your precious time to constructive comments on our manuscript titled “**Enabling Dynamic Multicolor Emission and Photo-Thermo-Mechanically Responsive Luminescence via Trace Mn²⁺ Doping in Self-Activated CaGa₄O₇** (NCOMMS-23-33295A) for *Nature Communications*. We sincerely appreciate your opinions and confirmation of our work. Accordingly, we have made the corresponding response and revision based on those comments.

Reviewer #1: The manuscript entitled “Enabling Dynamic Multicolor Emission and Photo-Thermo-Mechanically Responsive Luminescence via Trace Mn²⁺ Doping in Self-Activated CaGa₄O₇” written by Y. Tang et al. reports on the unconventional optical properties of CaGa₄O₇ doped with Mn²⁺. Phosphors were elaborated following a high temperature solid-state reaction, leading to micrometric powder. Prepared phosphors demonstrate intense luminescence with interesting unconventional properties: dynamic color change, thermal color change, thermally enhanced luminescence and mechanoluminescence. An almost complete set of optical characterization is proposed to describe these properties and theoretical calculations are coupled to additional characterization methods to unravel the origin of the observed optical properties. Finally, authors embedded the developed phosphor in a polymer matrix to demonstrate interesting anticounterfeiting perspectives and place their material into an applicative context. I personally think that the proposed work might interest readers from material science and phosphor communities. I therefore recommend this work to be published in Nature Communication but I would like to have the following queries resolved before.
[Authors’ Response]: Thanks for your positive comments and confirmation of our work. Your precious comments are highly significant for the revision of our manuscript. To address your kind comments, we have supplied detailed explanations. Here is our point-by-point response to the comments. We sincerely hope that our responses and revisions can fully satisfy your high requirements for publication.

1. In the beginning of the articles, authors tune Mn²⁺ concentration to vary phosphor color. As, latter on, it is demonstrated that the color varies with source characteristics (excitation time for instance), I would suggest to specify excitation characteristics in figure caption (Figure S2 here). This might be done systematically.

[Authors’ Response]: Thanks very much for your comments. We have added detailed information about the excitation characteristics in the caption of Supplementary Fig. 2 “The optical power density for photoexcitation was fixed at 0.5 W cm⁻². Note that these spectra were taken on the phosphors that have reached stable emission, when the PL intensity remains constant upon continuous photoexcitation.”

2. Do authors have an idea of the typical luminescence efficiency of developed phosphors? For instance, when doping content is varied I regret that PLQY is not discussed.

[Authors’ Response]: Thank you very much for your comments. We have performed the PLQY measurement and added regarding discussion in **Supplementary Fig. 4**.

Supplementary Fig. 4 | PLQY of the CaGa₄O₇:Mn²⁺ phosphors at room temperature. a-h, Absolute PLQY measurement of the Ca_{1-x}Ga₄O₇:xMn²⁺ (x = 0–10 × 10⁻⁴) phosphors. **i,** Mn²⁺ concentration-dependent PLQY of Ca_{1-x}Ga₄O₇:xMn²⁺. We noticed that the PLQY gradually decreased with increasing Mn²⁺ concentration from x = 0 to x = 2 × 10⁻⁴, while it gradually enhanced from x = 2 × 10⁻⁴ to x = 10 × 10⁻⁴. This is because the blue light emission of the CaGa₄O₇ host is burst by Mn²⁺ doping in the former stage, while the Mn²⁺ emission is continuously enhanced in the latter stage.

3. Various spectral evolution of CaGa₄O₇ and CaGa₄O₇:Mn²⁺ emission with time and temperature are shown all along the manuscript (Fig.3, Fig.4). I think that CIE diagrams can be placed here to ease readability.

[Authors' Response]: Thanks very much for your suggestions. We have added the CIE diagrams for various spectral evolution of CaGa₄O₇ and CaGa₄O₇:Mn²⁺ emission with time, power density and temperature. Considering the limitations of the layout, these additions are shown in **Supplementary Figs. 6b, 9, 11, 13 and 14.**

Supplementary Fig. 6 | Photoexcitation time-dependent multicolor emission of $\text{CaGa}_4\text{O}_7:\text{Mn}^{2+}$. b, Dependence of the CIE chromaticity coordinates on photoexcitation time (0.5–10 s).

Supplementary Fig. 9 | CIE chromaticity coordinates of the host emission from undoped CaGa_4O_7 with increasing photoexcitation time (0.25–7 s). The results show only minor changes in the color coordinates.

Supplementary Fig. 11 | Dependence of rate of emission color change on re-excitation intervals and heat treatment. **a**, Re-excitation interval of 60 s; **b**, Re-excitation interval of 300 s; **c**, Re-excitation interval of 900 s. **d**, Heat treatment (473 K, 10 s). These results illustrate that a shorter re-excitation interval leads to a faster evolution of the emission color, and a simple heat treatment can rapidly restore the evolutionary rate to the initial state when the phosphor was first photoexcited.

Supplementary Fig. 13 | Dependence of rate of emission color change on the optical power density. **a–c**, Standard CIE chromaticity graphs under different optical power densities (0.25, 0.5 and 0.75 W cm⁻², $\lambda_{\text{ex}} = 254$ nm). These results illustrate that a higher optical power density leads to a faster evolution of the emission color.

Supplementary Fig. 14 | Dependence of the CIE chromaticity coordinates of the PL spectra of $\text{CaGa}_4\text{O}_7:\text{Mn}^{2+}$ on temperature.

4. Authors discuss on the evolution of blue and yellow emissions kinetics with excitation power. I would like to see the profiles of this kinetics as it is done for the experiment with re-excitations.

[Authors' Response]: Thank you very much for your comments. In response to your suggestion, we have added profiles of the evolution of the blue and yellow emissions kinetics with excitation power, as shown in Figure 3j. “The transient PL images (Fig. 3j) show that as the optical power density increases, the locations of blue and yellow emissions remain unchanged, nevertheless the enhancement rate of blue emission and the decay rate of yellow emission become faster, which accelerates the evolution of the overall emission color.” It should be noted that the phosphors were heat-treated to evacuate the filled traps prior to each test with a different excitation power, as noted in the caption of Fig. 3j.

Fig. 3 | Dynamic multicolor evolution of photoluminescence at room temperature. j, Time-dependent transient PL images of $\text{CaGa}_4\text{O}_7:\text{Mn}^{2+}$ under different optical power densities. Note that the phosphors were heat-treated to evacuate the traps prior to each test.

5. Authors observed an increase in Mn^{2+} emission with temperature. Can authors comment on the evolution of the luminescence efficiency with temperature?

[Authors' Response]: Thank you very much for your comments. Currently, testing of variable temperature PLQYs remains a challenge. In response to your suggestion, we have estimated the PLQY values of Mn^{2+} emission with increasing temperature, as shown in Figure R1. This estimation is based on the tested PLQY of $\text{CaGa}_4\text{O}_7:\text{Mn}^{2+}$ (0.01 mol%) at room temperature, the percentage of Mn^{2+} emission obtained from the

split-peak fitting, and experimentally measured relative emission intensity of Mn^{2+} at elevated temperatures.

$$D_{\text{Estimated emission efficiency}} = A_{\text{PLQY@RT}} \times B_{\text{Mn}^{2+} \text{ emission percentage}} \times C_{\text{Relative intensity}}$$

The highest emission efficiency of 95.94% is estimated at 498 K, the strongest relative intensity. Notably, the intensity of the excitation peak was assumed not to change with increasing temperature in the estimation, which may bias the estimates high.

Figure R1. Estimated Mn^{2+} emission efficiency of $\text{CaGa}_4\text{O}_7:\text{Mn}^{2+}$ (0.01 mol%) with increasing temperature (298–623 K). a, Absolute PLQY measurements at room temperature. **b,** Percentage of Mn^{2+} emission obtained from the split-peak fitting. **c,** Experimentally measured relative emission intensity of Mn^{2+} at elevated temperatures. **d,** Estimated Mn^{2+} emission efficiency with increasing temperature.

6. A spectral shift of Mn^{2+} band is observed with temperature, associated with a thermally induced lattice deformation. Still a plateau can be observed from 300 to 400 K. How can authors explain this plateau? Please add some references.

[Authors' Response]: Thank you very much for your comments. From 300 K to 350 K, the enhancement of Mn^{2+} emission is slow and appears to be a plateau, but in fact, Mn^{2+} emission still increases with temperature (Fig. 4c). In contrast, from 350 K to 450 K, Mn^{2+} emission increases rapidly. These observations can be explained by the temperature-dependent distribution of the ThL peaks (Fig. 4e). The slower enhancement at 300-350 K is due to the fact that the ThL shoulder is weaker in this range when less energy is transferred from the traps to the Mn^{2+} ions. The rapid enhancement of Mn^{2+} emission thereafter is due to the appearance of a towering ThL peak at 350-425 K, where energy transfer from the traps to Mn^{2+} ions is greatly enhanced. Relevant references have been cited as follows.

48. Kim, Y. H. et al. A zero-thermal-quenching phosphor. *Nat. Mater.* **16**, 543–550 (2017).
49. Qiao, J. et al. Eu^{2+} site preferences in the mixed cation $\text{K}_2\text{BaCa}(\text{PO}_4)_2$ and thermally stable luminescence. *J. Am. Chem. Soc.* **140**, 9730–9736 (2018).
50. Tang, Z. et al. Highly efficient and ultralong afterglow emission with anti-thermal quenching from $\text{CsCdCl}_3:\text{Mn}$ perovskite single crystals. *Angew. Chem. Int. Ed.* **61**, e202210975 (2022).

7. To discuss on the origin of the blue luminescence from host material, authors investigated potential impurities and from simulations concluded that blue emission might originate from electron-hole recombination from impurity level (Si_{Ga^+} , Al_{Ga} , Zn_{Ca} or Ge_{Ga^+}) to V_{GaO^-} defect level. Is it possible to prove these conclusions? Can authors imagine intentionally doping CaGa_4O_7 matrices by Si, Al, Zn and Ge to reveal the genuine origin of the blue emission? If the real origin of the blue emission is found, can it be used to enhance it and tune the dynamic luminescence properties?

[Authors' Response]: Thanks very much for your comment. We have updated calculations and added relevant experiments. “We began by examining the defect states associated with the blue emission in CaGa_4O_7 , specifically considering unintentional impurity defects. This is because the intensity of the blue emission diminished when CaGa_4O_7 was prepared using Ga_2O_3 with higher purity (**Supplementary Fig. 19**). Inductively coupled plasma-mass spectrometer (ICP-MS) measurements of the prepared CaGa_4O_7 confirmed the presence of trace impurities (Supplementary Table S4). These results indicate that residual impurities present in the raw materials might contribute to the blue emission observed in the host material.” “Then, we systematically reveal the electronic structures of different intrinsic defects in the host materials. For the impurities, we notice that only 3d orbitals of Cu impurity induce an empty state at $E_V+2.61$ eV, which is able to realize the blue emission with a wavelength of 475 nm (**Supplementary Fig. 20a–d**). In comparison, all the other impurities cannot improve the blue emissions since their states are deep in the VB and CB.”

Based on your suggestion, we then synthesized CaGa_4O_7 matrices intentionally doped with trace (100 ppm) Cu, Zn, Al, Si, and Ge impurities. PL tests demonstrated that the introduction of impurities did not alter the profile and location of the excitation and emission peaks, but changed the intensity of these peaks in CaGa_4O_7 (**Supplementary Fig. 21**). However, a noticeable change was observed in the intensity of these peaks. Specifically, Cu dopants resulted in a modest 0.2% increase in host emission, while Zn, Al, Si, and Ge dopants led to a reduction in emission intensity within the range of 0.6% to 40.1%. These experimental findings consistently align with theoretical calculations. These results underscore that even trace impurities have the capability to finely modulate the luminescence performance of CaGa_4O_7 . Extensive sample preparation and systematic studies will be further carried out to optimize both the selection of elemental species and the appropriate doping amount.

Supplementary Fig. 19 | PLE and PL spectra of undoped CaGa_4O_7 prepared from Ga_2O_3 raw materials with different purities (99.99% and 99.999%). a, PLE spectra ($\lambda_{\text{em}} = 474$ nm). b, PL spectra ($\lambda_{\text{ex}} = 254$ nm). The results show that the intensity of blue emission diminishes when higher purity of Ga_2O_3 is used, suggesting that the unintentionally doped impurities in the material might contribute to the blue emission of the host.

Supplementary Fig. 21 | PL properties of CaGa_4O_7 intentionally doped by Cu, Zn, Al, Si, and Ge (0.01 mol%) impurities. **a**, Excitation spectra ($\lambda_{\text{em}} = 474$ nm). **b**, Emission spectra ($\lambda_{\text{ex}} = 256$ nm). **c**, Relative emission intensity of doped samples compared to undoped CaGa_4O_7 . PL tests demonstrated that the introduction of impurities did not alter the profile and location of the excitation and emission peaks, but changed the intensity of these peaks in CaGa_4O_7 . However, a noticeable change was observed in the intensity of these peaks. Specifically, Cu dopants resulted in a modest 0.2% increase in host emission, while Zn, Al, Si, and Ge dopants led to a reduction in emission intensity within the range of 0.6% to 40.1%. These experimental findings consistently align with theoretical calculations. These results underscore that even trace impurities have the capability to finely modulate the luminescence performance of CaGa_4O_7 . Extensive sample preparation and systematic studies will be further carried out to optimize both the selection of elemental species and the appropriate doping amount.

8. I think that PL decays of the blue and yellow emission can bring interesting information when discussing the luminescence mechanisms.

[Authors' Response]: Thank you very much for your comments. According to your kind suggestion, we have tested the PL decay curves of the blue and yellow emissions of the $\text{Ca}_{1-x}\text{Ga}_4\text{O}_7:\text{xMn}^{2+}$ ($x = 0-10 \times 10^{-4}$) phosphors (**Supplementary Fig. 3**) and added the relevant discussion in the figure caption.

Supplementary Fig. 3 | PL decay curves of the blue and yellow emissions of the $\text{Ca}_{1-x}\text{Ga}_4\text{O}_7:\text{xMn}^{2+}$ ($x = 0-10 \times 10^{-4}$) phosphors at room temperature. **a**, Blue emission ($\lambda_{\text{ex}} = 260$ nm, $\lambda_{\text{em}} = 474$ nm). **b**, Yellow emission ($\lambda_{\text{ex}} = 254$ nm, $\lambda_{\text{em}} = 579$ nm). Number of photons collected: 10000. The decay curves of blue emission were fitted by a bi-exponential function. Two lifetimes were obtained by fitting each curve, the first being the response time of the instrument (here a microsecond lamp was used as the excitation source) and the second being the lifetime of the tested sample, as shown in Supplementary Fig. 3a. The decay curves of yellow emission were fitted by a monoexponential function. These results show that both the lifetime of blue emission and the lifetime of yellow emission are gradually shortened with increasing Mn^{2+} doping concentration. The former shortening may be due to the energy competition of the doped Mn^{2+} ions,^{S1} while the latter shortening is possibly caused by the intensified non-radiative transitions of Mn^{2+} ions with the increase of Mn^{2+} doping concentration.^{S2} To the best of our knowledge, no blue emission lifetime has been reported in

CaGa₄O₇. As a comparison, the magnitude of the decay time of host emission in CaGa₄O₇ is much longer than the reported values (1.8 μs and 2.511 μs) for self-activated MgGa₂O₄ phosphor, which also has blue emission from host.^{S3, S4} The magnitude of the decay times of Mn²⁺ emission is similar to the previously reported values (10.927–7.083 ms) in Ca_{1-x}Ga₄O₇:xMn²⁺ (x = 0.002–0.1),^{S5} suggests that the luminescence of Mn²⁺ in CaGa₄O₇ is a spin-forbidden transition.

Updated supplementary reference:

S1. Piotrowski, W. M. et al. Positive luminescence thermal coefficient of Mn²⁺ ions for highly sensitive luminescence thermometry. *Chem. Eng. J.* **464**, 142492 (2023).

S2. Xu, H. et al. 2D perovskite Mn²⁺-doped Cs₂CdBr₂Cl₂ scintillator for low-dose high-resolution X-ray imaging. *Adv. Mater.* **35**, 2300136 (2023).

S3. Liu, Z. et al. Luminescence of native defects in MgGa₂O₄. *J. Electrochem. Soc.* **156**, H43–H46 (2009).

S4. Jiang, B. et al. A self-activated MgGa₂O₄ for persistent luminescence phosphor. *J. Appl. Phys.* **124**, 063101 (2018).

S5. Zheng, W. et al. Crystal field modulation-control, bandgap engineering and shallow/deep traps tailoring-guided design of a color-tunable long-persistent phosphor (Ca,Sr)Ga₄O₇:Mn²⁺,Bi³⁺. *Dalton Trans.* **48**, 253–265 (2019).

9. Why can the traps recombine only on Mn²⁺ centers and not on defects responsible of blue emission? Is it due to a local charge recombination? Can simulations prove that oxygen vacancies are more stable in the vicinity of Mn²⁺ impurity? Did authors record ThL of undoped samples? If yes, electrons might not go through the CB as represented in the scheme 6J.

[Authors' Response]: Thank you very much for your comments. Following your kind suggestion, we have demonstrated that the oxygen vacancies become more stable with the vicinity of Mn²⁺ based on additional DFT calculations (**Fig. 6i**). More importantly, the formation energies of defects also decrease after the Mn are introduced in the system, supporting that defects can be stable even near Mn²⁺ (**Fig. 6g-h**). As a result, the local charge recombination occurs between the traps and Mn²⁺ centers, rather than the traps and the defects leading to blue emission. It can be confirmed by the results of the ThL tests. Undoped CaGa₄O₇ shows a ThL spectrum with blue emission, nevertheless, in the ThL spectra of CaGa₄O₇:Mn²⁺, there is only Mn²⁺ emission and no blue emission (inset of Fig. 4e and Figure R2). Based on your expert opinion, we have modified the electron transport path between the trap and Mn²⁺ to a tunneling effect.

Fig. 6g-i. The formation energies of defects in (g) CaGa₄O₇ and (h) CaGa₄O₇:Mn. (i) The formation energy changes of oxygen vacancies with different distances to Mn²⁺.

Figure R2. ThL curves and spectra of CaGa_4O_7 and $\text{CaGa}_4\text{O}_7:\text{Mn}^{2+}$. **a**, ThL curves. **b**, ThL spectra.

10. Can authors develop on the explanation of the decrease of the yellow (Mn^{2+}) emission with excitation time? I personally find this part a little confusing. It is said that filling of traps (Vo^{2+} , Vo^+) compete with yellow emission, leading to the decrease of yellow emission. When traps are fully filled one can expect that the competition between the two process will no longer occur, leading into more intense yellow emission, a typical for persistent phosphors. How can observed results be placed in the context of persistent phosphors loading curves (10.1002/adpr.202100179, 10.1364/OME.6.000844)? Is it possible that available Mn^{2+} ions diminish through photo-oxidation due to electron trapping?

[Authors' Response]: Thanks very much for your comments. The yellow emission decreases with excitation time, which can be explained by the energy transfer from Mn^{2+} ions to the color center. When the $\text{CaGa}_4\text{O}_7:\text{Mn}^{2+}$ phosphor is exposed to UV photoexcitation, a photochromic phenomenon occurs, indicating the formation of a color center. **Figure 7a** shows that the yellow emission of Mn^{2+} (500–700 nm) lies within the absorption range of the color center (300–700 nm). Due to the overlap between the emission band and the absorption band, the energy of the luminescent center (Mn^{2+}) is partially transferred to the color centers to release the electrons trapped by the oxygen vacancies during the photochromic process. As a result, a decrease in Mn^{2+} emission is observed in $\text{CaGa}_4\text{O}_7:\text{Mn}^{2+}$. Such phenomena have been reported in many phosphors (e.g., $\text{BaMgSiO}_4:\text{M}$ ($\text{M} = \text{Eu}^{2+}$, Ce^{3+} , Mn^{2+} , Nd^{3+}) and $\text{Na}_{0.5}\text{Bi}_{4.5}\text{Ti}_4\text{O}_{15}:\text{RE}$ ($\text{RE} = \text{Sm}$, Pr , Er)) and the mechanisms of energy transfer have been accepted generally (Refs. R1,60,61).

Fig. 7 | Optical transition and energy transport mediated by defects. **a**, Diffuse reflectance spectra of $\text{CaGa}_4\text{O}_7:\text{Mn}^{2+}$ before and after photoexcitation and the PL spectrum of Mn^{2+} . The inset is a photograph of a $\text{CaGa}_4\text{O}_7:\text{Mn}^{2+}$ disk photoexcited on the right half.

The loading curves of the traditional persistent phosphors (e.g., SrAl₂O₄:Eu,Dy and Sr₂MgSi₂O₇:Eu,Dy) at room temperature rise monotonically with photoexcitation time, reflecting the charging process of the traps. The luminescent intensity saturates when the trapping and de-trapping of the traps reaches dynamic equilibrium. However, the emission decay phenomenon in this work does not contradict the abovementioned loading curves. Conventional persistent phosphors do not form color centers when the traps are charged and thus do not absorb the energy corresponding to the characteristic emission of the luminescent centers. In contrast, in this work, CaGa₄O₇:Mn²⁺ phosphors form color centers when charged under photoexcitation, and the color centers can absorb the energy of Mn²⁺ ions, thereby attenuating the Mn²⁺ emission. The Mn²⁺ emission is stabilized when the concentration of the formed color center is saturated and the trapping and de-trapping rates of the color center are in dynamic equilibrium.

In the present work, we confirmed the absence of photo-oxidation of Mn²⁺ ions from two points. First, no emission of high-valent Mn ions was detected. Fig. 3b and Supplementary Fig. 6a show that the emission intensity of Mn²⁺ ions decays by more than 60% within 10 seconds. If such a large attenuation is due to the photo-oxidation of Mn²⁺ ions, then a large percentage of high-valent Mn ions should be formed and their characteristic emission should be detected. However, in the PL spectra of the phosphors after photoexcitation, there is still only Mn²⁺ emission, and no characteristic emission of any other Mn ions was detected, such as Mn³⁺ (Refs. 55 and R2), Mn⁴⁺ (Ref. 56), and Mn⁵⁺ (Ref. 57). Second, the experimental results of the XPS spectra showed that the chemical state of Mn ions did not change before and after photoexcitation (**Supplementary Fig. 25**). It is well known that XPS parameters can be considered to determine the chemical states of Mn ions. If the oxidation state of Mn increases, the binding energy of the Mn2p_{3/2} peak will increase, while the Mn3s multiplet splitting value (defined as the energy difference between two prominent peaks) will decrease (Refs. R3,58,59). However, in the present study, the binding energy of Mn2p_{3/2} did not change, nor did the Mn3s multiplet splitting value (5.4 eV), indicating that photoexcitation does not modify the chemical state of the Mn²⁺ ions.

Supplementary Fig. 25 | XPS spectra of Ca_{1-x}Ga₄O₇:xMn²⁺ (x = 1×10⁻⁴) before and after photoexcitation. a, Mn2p spectra. b, Mn3s spectra.

It should be noted that, although photo-induced oxidation of Mn²⁺ ions has been reported in the literature, these irradiations generally belong to high doses of high-energy rays and high-intensity UV light, e.g., γ -rays (1.6 kGy/h; Ref. R4), X-rays (55 kV, 30 mA, 20 min; Ref. R5), UV irradiation (200 W/cm², 20 min; Ref. R6). In

comparison, in our work, the intensity of UV photoexcitation is very low, only 0.25–0.75 W/cm², and the photoexcitation time is very fast, only a few seconds. Such low intensity and fast UV irradiation may not be sufficient to oxidize Mn²⁺ ions.

References used for responses:

R1. Zhang, Q. et al. Tunable luminescence contrast of Na_{0.5}Bi_{4.5}Ti₄O₁₅:Re (Re = Sm, Pr, Er) photochromics by controlling the excitation energy of luminescent centers. *ACS Appl. Mater. Interfaces* **8**, 34581–34589 (2016).

R2. Marciniak, L. and Trejgisa, K. Luminescence lifetime thermometry with Mn³⁺–Mn⁴⁺ co-doped nanocrystals. *J. Mater. Chem. C* **6**, 7092–7100 (2018).

R3. Chando, P. A. et al. Exploring calcium manganese oxide as a promising cathode material for calcium-ion batteries. *Chem. Mater.* **35**, 8371–8381 (2023).

R4. Zhydachevskii, Ya. et al. Optical observation of the recharging processes of manganese ions in YAlO₃:Mn crystals under radiation and thermal treatment. *J. Phys.: Condens. Matter* **18**, 5389–5403 (2006).

R5. Romet, I. et al. Recombination luminescence and EPR of Mn doped Li₂B₄O₇ single crystals. *Opt. Mater.* **70**, 184–193 (2017).

R6. Shin, H. et al. Photochromic effect in stoichiometric LiNbO₃:Fe,Mn. *Jpn. J. Appl. Phys.* **43**, 7504–7507 (2004).

Updated references

55. Jahanbazi, F. et al. Tb³⁺,Mn³⁺ co-doped La₂Zr₂O₇ nanoparticles for self-referencing optical thermometry. *J. Lumin.* **240**, 118412 (2021).

56. Zhu, H. et al. Highly efficient non-rare-earth red emitting phosphor for warm white light-emitting diodes. *Nat. Commun.* **5**, 4312 (2014).

57. Piotrowski, W. M. et al. Mn⁵⁺ lifetime-based thermal imaging in the optical transparency windows through skin-mimicking tissue phantom. *Adv. Opt. Mater.* **11**, 2202366 (2023).

58. Yang, S. et al. Effect of proton transfer on electrocatalytic water oxidation by manganese phosphates. *Angew. Chem. Int. Ed.* **62**, e202215594 (2023).

59. Ilton, E. S. et al. XPS determination of Mn oxidation states in Mn (hydr)oxides. *Appl. Surf. Sci.* **366**, 475–485 (2016).

60. Yang, Z. et al. Designing photochromic materials with large luminescence modulation and strong photochromic efficiency for dual-mode rewritable optical storage. *Adv. Opt. Mater.* **9**, 2100669 (2021).

61. Tang, W. et al. Designing photochromic materials with high photochromic contrast and large luminescence modulation for hand-rewritable information displays and dual-mode optical storage. *Chem. Eng. J.* **435**, 134670 (2022).

11. A tilt in the band structure is observed under mechanical stimulation. Can Mn²⁺ emission band (⁴T₁) energy be shifted through mechanical stimulation too?

[Authors' Response]: Thank you very much for your comments. Following your kind suggestion, we have carried out additional theoretical calculations on the responses of band structures of CaGa₄O₇:Mn²⁺ under mechanical stimulations. To simulate the external mechanical stimulations, we have considered two different situations including both strain and stress. For the applied strain, we have applied 1%, 3%, and 5% strain on the (001) direction to simulate lattice deformation under mechanical stimulations. As shown in **Supplementary Fig. S24a-d**, we have noticed that the corresponding band structures of CaGa₄O₇ show an evident tilt trend, where both valence band maximum (VBM) and conduction band minimum (CBM) gradually downshift as the applied strain increases. Although the CBM and VBM display the shifting trend, we notice that the Mn²⁺ has remained in the same position within the bandgap with applying strain, supporting that the emission band (⁴T₁) energy remains stable under the mechanical stimulations. On the other side, we have evaluated the bandstructures of CaGa₄O₇:Mn²⁺

under applying stress of 10 GPa, 30 GPa and 50 GPa (**Supplementary Fig. S24e-h**). Notably, the corresponding band structures also display a similar trend to the applying strain, where the bandstructure of CaGa_4O_7 gradually shifts with the increasing stress while the states of Mn^{2+} are well pinned within the bandgap. These results further confirm that Mn^{2+} emission band (${}^4\text{T}_1$) energy positions are not shifted. The tilt of bandstructure leads to the reduced energy barrier between Mn^{2+} and defect traps.

Supplementary Fig. 24 | The bandstructure of CaGa_4O_7 : Mn with external strain and stress. Strain: (a) 0%, (b) 1%, (c) 3% and (d) 5% on the (001) direction. Stress: (e) 0 GPa, (f) 10 GPa, (g) 30 GPa, and (h) 50 GPa.

12. In the final “applications” part, the developed phosphor is embedded in silicone rubber, chosen for its thermal, mechanical as well as optical transparency characteristics. However, the proposed flexible matrix demonstrates strong absorption in the UV range below 300 nm -see Fig. S22-. Is it the best matrix knowing a 254 nm UV lamp is used to excite the phosphor? The optical power density felt by the phosphor will therefore be different compared to the powdered phosphor. Do authors think that this is a potential application issue knowing that the dynamic optical properties of the developed phosphor depends on the source intensity -see Fig. S6-? Also, authors should be careful when mentioning the potentiality of optical sensing of ambient temperature. How is the silicone rubber matrix influencing the thermal conductivity of the sensor and the emission ratio (I_{579}/I_{474})? Indeed, its absorption at 579 nm and 474 nm is different. I would therefore like to see similar “Temperature-dependent luminescent intensity ratio and relative sensitivity” conducted on $\text{CaGa}_4\text{O}_7:\text{Mn}^{2+}$ devoid of polymer matrix.

[Authors’ Response]: Thank you very much for your comments. Based on your suggestion, we have added relevant experiments and explanations. In this work, the silicone rubber was chosen as the elastomeric matrix due to its exceptional combination of thermal-mechanical-optical properties, including thermal stability (up to 573 K, **Supplementary Fig. 31**), mechanical stretchability (over 300%, **Supplementary Fig. 32**) and optical transparency (over 88% and flat transmittance in the 400-700 nm range,

Supplementary Fig. 33). The silicone rubber used does not have the highest transmittance at 254 nm, but it has high thermal stability and high stretchability. The experimentally observed rate of color change of the composite film was essentially the same as that of the powder, due to the fact that the phosphor was uniformly dispersed in the thin polymer, which has a very limited effect on the optical power density felt by the phosphor. We updated the optical transmittance spectrum of the used silicone rubber (Supplementary Fig. 33), where the film previously tested may have been contaminated in use. As can be seen from the Supplementary Fig. 33, the curve is very flat in the 400–800 nm range, where the transmittance exceeds 88%. In particular, the transmittances at 579 nm (89.8%) and 474 nm (89.7%) are almost identical. Therefore, the change in the emission ratio (I_{579}/I_{474}) due to the silicone rubber used is negligible. We have supplemented the temperature-dependent luminescent intensity ratio and relative sensitivity of $\text{CaGa}_4\text{O}_7:\text{Mn}^{2+}$ phosphors devoid of polymer matrix (**Supplementary Fig. 35**). We compared the results related to composite film and $\text{CaGa}_4\text{O}_7:\text{Mn}^{2+}$ phosphor. It was observed that the Sr values of the composite film are lower than those of the phosphor, which is due to the lower thermal conductivity of the polymer matrix. Here, we showed conceptual demonstrations for potential applications. For future practical applications, the effects of different polymer types, thicknesses, thermal conductivity and other factors need to be investigated in detail.

Supplementary Fig. 31 | Photographs of five $\text{CaGa}_4\text{O}_7:\text{Mn}^{2+}$ /polymer composite films at different temperatures (298–573 K). The silicone rubber-based film remained flat with increasing temperature, while the other composite film curled above 473 K. These results indicate that the silicone rubber-based composite film has better thermal stability than the other composite films. The size of composite films is $40 \times 40 \times 1 \text{ mm}^3$. Note that five organic polymers were used to prepare these $\text{CaGa}_4\text{O}_7:\text{Mn}^{2+}$ /polymer composite films, including silicone rubber (Xinbang Chemical Technology), PDMS (Polydimethylsiloxane, Dow Corning Shanghai), RTVS605 (Dimethylvinylated and trimethylated silica + Hydrogen-terminated vinyl silicone polymer + Vinyl silicone polymer + Polydimethylsiloxane, Hasuncast Industries), HL1028 (Polydimethylsiloxane + Vinyl silicone oil, Shenzhen Huasheng Tongchuang Technology), TY866 (Silica powder + Thermal conductive powder + Polydimethylsiloxane, Dongguan Tianyu Composite Materials). The phosphor microparticles were incorporated into the organic polymers at a weight ratio of 2:1. The curing conditions for these composite films were as follows: silicone rubber-based film cured at 333 K for 10 minutes, PDMS-based films cured at 353 K for 2 hours, RTVS605-based films cured at 373 K for 30 minutes, HL1028-

based films cured at 353 K for 15 minutes, and TY866-based films cured at 353 K for 360 minutes.

Supplementary Fig. 32 | Load-elongation curves of five $\text{CaGa}_4\text{O}_7:\text{Mn}^{2+}$ /polymer composite films. It indicates higher stretchability for the silicone rubber-based composite.

Supplementary Fig. 33 | Optical transmittance spectrum of organic films synthesized from different materials. The inset shows the optical photographs of a transparent silicone rubber film ($60 \times 25 \times 1 \text{ mm}^3$).

Supplementary Fig. 35 | Temperature-dependent luminescent intensity ratio (R) and relative sensitivity (S_r). **a**, $\text{CaGa}_4\text{O}_7:\text{Mn}^{2+}/\text{PDMS}$ composite film. $R(I_{579}/I_{474}) = 34839.57\exp(-2860.73/T) - 7.27$. $S_r = \frac{1}{R} \frac{dR}{dT}$.¹⁹ $S_r = 2.74\% \text{ K}^{-1}$ @ 323 K; $1.28\% \text{ K}^{-1}$ @ 473 K. **b**, $\text{CaGa}_4\text{O}_7:\text{Mn}^{2+}$ phosphor devoid of polymer matrix. $R(I_{579}/I_{474}) = 57969.04\exp(-3066.26/T) - 6.55$. $S_r = 3.09\% \text{ K}^{-1}$ @ 323 K; $1.44\% \text{ K}^{-1}$ @ 473 K. The S_r values of the composite film are lower than those of the phosphor, which is due to the lower thermal conductivity of the polymer matrix.

Reviewer #2: The manuscript "Enabling Dynamic Multicolor Emission and Photo-Thermo-Mechanically Responsive Luminescence via Trace Mn²⁺ Doping in Self-Activated CaGa₄O₇" discusses the optical properties of Mn-doped gallate crystals. The authors show that these crystals exhibit unique time-dependent dual luminescence, temperature-dependent emission, and mechanoluminescence, which could have valuable applications in the detector and anti-counterfeiting industry. The experimental data presented in the manuscript supports the authors' claims. However, the data quality and analysis require significant improvements before definite recommendation for publication.

The major issues are listed below:

[Authors' Response]: Thank you very much for your positive comments and validation of our work. We sincerely appreciate the time and effort you have dedicated to reviewing our research. We have carefully considered your suggestions and have made revisions accordingly. We sincerely hope that our responses and revisions meet the high standards you expect for publication.

1. The authors should include an analysis of the presence of Mn⁴⁺ in their experimental results, as different oxidation states of Mn have different optical properties. Additionally, the statement "Photoluminescent excitation (PLE) spectra, monitored at different emission wavelengths ranging from 400 to 625 nm, exhibit identical shapes and features, with an excitation peak observed at 256 nm (Fig. 3c). This result indicates that the blue and yellow emissions arise from the relaxation of the same excited state" lacks precision. While the VB-CB transition might be dominant under the above bandgap excitation, a PLE spectrum extended to the visible range should show the atomic level transitions of the defects and the Mn dopant. The PLE peaks related to such transitions might be significantly weaker than the direct band gap transition of the host due to the forbidden transitions between the electron levels of the Mn ions.

[Authors' Response]: Thanks very much for your valuable comments. Following your suggestion, we have added an analysis of the experimental results to show the absence of Mn⁴⁺ as well as Mn³⁺ and Mn⁵⁺ ions in the materials. "To investigate the gradual weakening of the yellow emission upon photoexcitation, we first considered the possibility of photo-oxidation of Mn²⁺ ions. The kinetics of the PL spectra show the emission intensity of Mn²⁺ ions decays by more than 66% within 15 seconds (Fig. 3h,3j). If such a large attenuation was due to the photo-oxidation of Mn²⁺ ions, a large percentage of high-valent Mn ions (e.g., Mn³⁺, Mn⁴⁺ and Mn⁵⁺) would be formed. However, the investigation of PL spectra shows that only Mn²⁺ emission was observed before and after photoexcitation, and no emission of other Mn ions was detected (Fig. 3b and Supplementary Fig. 2).⁵⁵⁻⁵⁷ XPS spectra show that no change in the binding energy of Mn2p_{3/2} or in the Mn3s multiplet splitting value (5.4 eV) before and after photoexcitation (Supplementary Fig. S25). These experimental results confirm that photoexcitation did not modify the chemical state of the Mn²⁺ ions."^{58,59}

Updated references:

55. Jahanbazi, F. et al. Tb³⁺, Mn³⁺ co-doped La₂Zr₂O₇ nanoparticles for self-referencing optical thermometry. *J. Lumin.* **240**, 118412 (2021).

56. Zhu, H. et al. Highly efficient non-rare-earth red emitting phosphor for warm white light-emitting diodes. *Nat. Commun.* **5**, 4312 (2014).

57. Piotrowski, W. M. et al. Mn⁵⁺ lifetime-based thermal imaging in the optical transparency windows through skin-mimicking tissue phantom. *Adv. Opt. Mater.* **11**, 2202366 (2023).

58. Yang, S. et al. Effect of proton transfer on electrocatalytic water oxidation by manganese phosphates. *Angew. Chem. Int. Ed.* **62**, e202215594 (2023).

59. Ilton, E. S. et al. XPS determination of Mn oxidation states in Mn (hydr)oxides. *Appl. Surf. Sci.* **366**, 475–485 (2016).

We have also added the extended PLE spectrum to show the transition levels of the Mn^{2+} ions (**Supplementary Fig. 7**). The enlarged spectrum (375–550 nm) shows the transitions of Mn^{2+} ions from the ground state ${}^6\text{A}_1({}^6\text{S})$ to the excited states ${}^4\text{A}_1({}^4\text{G})$, ${}^4\text{E}({}^4\text{G})$ (409 nm), ${}^4\text{T}_2({}^4\text{G})$ (510 nm) and ${}^4\text{T}_1({}^4\text{G})$ (522 nm) and do not show any significant transition of the defects and the high-valence Mn ions. Moreover, according to your comment, we have revised the relevant statement. “Photoluminescent excitation (PLE) spectra, monitored at different emission wavelengths ranging from 400 to 625 nm, exhibit identical shapes and features, with an intense excitation peak observed at 256 nm (Fig. 3c). Diffuse reflectance spectra demonstrate that the fixed excitation peak correlates with the host's absorption band (Supplementary Fig. 8), suggesting that the excitation processes for both host emission and Mn^{2+} emission in $\text{CaGa}_4\text{O}_7:\text{Mn}^{2+}$ are dominated by the transition from the valence band to the conduction band⁴⁴.”

Supplementary Fig. 7 | PL excitation spectrum of $\text{Ca}_{1-x}\text{Ga}_4\text{O}_7:x\text{Mn}^{2+}$ with $x = 1 \times 10^{-4}$ ($\lambda_{\text{em}} = 579 \text{ nm}$). The enlarged spectrum (375–550 nm) shows the transitions of Mn^{2+} ions from the ground state ${}^6\text{A}_1({}^6\text{S})$ to the excited states ${}^4\text{A}_1({}^4\text{G})$, ${}^4\text{E}({}^4\text{G})$ (409 nm), ${}^4\text{T}_2({}^4\text{G})$ (510 nm) and ${}^4\text{T}_1({}^4\text{G})$ (522 nm).

2. The characterization of the material's structure must include high-quality data. Figure 2 displays a high-resolution transmission electron microscopy (HR-TEM) image of the crystal edge and selected area electron diffraction (SEAD), but the data is challenging to comprehend. There is no information on how the sample preparation for HR-TEM was carried out, whether it was a focused ion beam (FIB) slice, or a small crystal found under the TEM.

[Authors' Response]: Thanks very much for your professional comments. We have added information on how the sample preparation for HR-TEM in the section of Method. “For the preparation of TEM samples, the phosphors were dispersed in ethanol, then dropped on a copper grid and dried on a hot plate (423 K), and tested on small crystals.” Furthermore, we have revised the typography of the labeled information in the HRTEM image (inset of **Fig. 2a**), and retested the SAED mode to get an image with higher quality (as shown in **Fig. 2b**).

Fig. 2 | Structural characterizations of as-synthesized $\text{CaGa}_4\text{O}_7:\text{Mn}^{2+}$. **a**, HRTEM image displaying lattice fringes in d-spacing. **b**, SAED pattern presenting the view down of the $[\bar{1}\bar{3}0]$ projection. **c**, HAADF-STEM image and corresponding compositional analysis of an individual particle. **d**, XRD pattern with the calculated profile obtained from Rietveld refinement.

3. In regard to high-angle annular dark-field scanning transmission electron microscopy (HAADF-STEM), the elemental distribution is not uniform and lacks explanation. The subfigure labeled "C" next to the STEM image is slightly confusing next to the elemental mapping. The X-ray diffraction (XRD) in the supplemental information (SI) shows a smaller range, and considering that gallates exist in different compositions, the XRD and quality factor of the Rietveld fitting should be reported.

[Authors' Response]: Thanks very much for your comments. Following your kind suggestion, we have retested the HAADF-STEM image on a single particle, demonstrating a uniform distribution of elements. The icon "c" has been placed outside the image to avoid misunderstanding. These modifications are shown in Figure 1c. Furthermore, according to your comments, we expanded the XRD display range to $5^\circ \leq 2\theta \leq 85^\circ$, confirming the crystallization of the resulting phosphors as a single phase. We also performed Rietveld refinement of the crystal structure on all the prepared $\text{Ca}_{1-x}\text{Ga}_4\text{O}_7:x\text{Mn}^{2+}$ ($x = 0, 0.2 \times 10^{-4}, 1 \times 10^{-4}, 2 \times 10^{-4}, 3 \times 10^{-4}, 4 \times 10^{-4}, 6 \times 10^{-4},$ and 10×10^{-4}) phosphors. The calculated results show a high quality factor. These modifications and additions are shown in **Supplementary Fig. 1a** and **Supplementary Fig. 4**, respectively.

Supplementary Fig. 1 | Characterization of the structure, morphology and elements of $\text{CaGa}_4\text{O}_7:\text{Mn}^{2+}$. (a) Powder XRD patterns of $\text{Ca}_{1-x}\text{Ga}_4\text{O}_7:x\text{Mn}^{2+}$ ($x = 0, 0.2 \times 10^{-4}, 1 \times 10^{-4}, 2 \times 10^{-4}, 3 \times 10^{-4}, 4 \times 10^{-4}, 6 \times 10^{-4},$ and 10×10^{-4}), demonstrating the synthesized $\text{CaGa}_4\text{O}_7:\text{Mn}^{2+}$ series as a single phase. (b) SEM image and (c) TEM-EDS spectrum of the synthesized $\text{Ca}_{1-x}\text{Ga}_4\text{O}_7:x\text{Mn}^{2+}$ ($x = 1 \times 10^{-4}$) pellet. The Cu element in the TEM-EDS spectrum arises from the copper mesh used in the TEM test.

Supplementary Fig. 4 | Rietveld refined XRD profiles and calculated parameters of $\text{CaGa}_4\text{O}_7:\text{Mn}^{2+}$. a–h, Rietveld refinement of XRD patterns for $\text{Ca}_{1-x}\text{Ga}_4\text{O}_7:x\text{Mn}^{2+}$. The calculated crystallographic structural parameters are summarized in Supplementary Table 2. i, Refined volume of the unit cell for $\text{Ca}_{1-x}\text{Ga}_4\text{O}_7:x\text{Mn}^{2+}$. The inset displays the crystal structure of CaGa_4O_7 . The monoclinic $C2/c$ (No. 15) space group is adopted by CaGa_4O_7 , and the crystal structure comprises two inequivalent Ga^{3+} sites and one Ca^{2+} site.⁶ Ca^{2+} is bonded to five O^{2-} atoms to form CaO_5 trigonal bipyramids that share corners with ten GaO_4 tetrahedra.

In theory, Mn^{2+} dopants are expected to occupy the Ca^{2+} sites due to the close radii and the crystal field effect. The radius percentage difference (D_r) values between Mn^{2+} (0.83 \AA , coordination number (CN)=6) and Ca^{2+} (1 \AA , CN=6) and between Mn^{2+} (0.66 \AA , CN=4) and Ga^{3+} (0.47 \AA , CN=4) are 17% and 40%, respectively.^{7,8} The calculation results indicate that the Mn^{2+} dopants are prone to occupy the Ca^{2+} sites because an acceptable D_r value cannot exceed 30%. Note that since there is no available radius data for Ca^{2+} with CN=5, data of CN=6 for Ca^{2+} and Mn^{2+} are used as reasonable approximations. Rietveld refinement analysis revealed a gradual decrease in the volume of the unit cell as the Mn^{2+} doping concentration increased, confirming the successful substitution of smaller Mn^{2+} ions for the Ca^{2+} sites. Moreover, Mn^{2+} in a five-coordinated environment emits yellow light,⁹ while Mn^{2+} in a tetra-coordinated environment emits green light instead of yellow light,¹⁰ which further confirms that Mn^{2+} dopants occupy the five-coordinated Ca^{2+} sites.

4. Although the experimental data is relatively well documented, the explanation of the physical process seems more speculative than experimentally supported. The $4T_1 \rightarrow 6A_1$ transition is parity and spin forbidden, meaning that any other relaxation

path is more likely. The UV excitation enables the excited electrons from the Mn ground state to be delocalized in the conduction band (CB) and locate a non-forbidden relaxation process. This process could change the Mn ion's oxidation (or charge) state, reducing the photoluminescence (PL) intensity. The electron paramagnetic resonance (EPR) and photo-EPR data reveal the generation of EPR active defects upon illumination, which may be oxygen vacancies in some of their charge states. Both the oxygen vacancy and the change of the oxidation state of Mn dopants change the color of metallic oxides. We strongly encourage the authors to review the description of the physical process written in the manuscript to explain the temporal and other changes in the optical properties. A more detailed EPR and photoluminescence excitation (PLE) study can answer most questions. If oxygen vacancies are generated upon UV excitation, those vacancies are likely to be permanent, and the EPR signal of the vacancy should evolve and show the same intensity in the dark. The change of the Mn oxidation state can also be followed by EPR and PLE. However, if the signal increase in photo-EPR is due to a charging effect, the signal should decrease rapidly in the dark.

[Authors' Response]: Thanks very much for your professional comments. Following your suggestion, we have added analysis and experimental results to identify the absence of photo-oxidation of Mn^{2+} ions. "To investigate the gradual weakening of the yellow emission upon photoexcitation, we first considered the possibility of photo-oxidation of Mn^{2+} ions. The kinetics of the PL spectra show the emission intensity of Mn^{2+} ions decays by more than 66% within 15 seconds (Fig. 3h,3j). If such a large attenuation was due to the photo-oxidation of Mn^{2+} ions, a large percentage of high-valent Mn ions (e.g., Mn^{3+} , Mn^{4+} and Mn^{5+}) would be formed. However, the investigation of PL spectra shows that only Mn^{2+} emission was observed before and after photoexcitation, and no emission of other Mn ions was detected (Fig. 3b and Supplementary Fig. 2)⁵⁵⁻⁵⁷. XPS spectra show no change in the binding energy of $\text{Mn}2p_{3/2}$ or in the $\text{Mn}3s$ multiplet splitting value (5.4 eV) before and after photoexcitation (Supplementary Fig. S25). These experimental results confirm that photoexcitation did not modify the chemical state of the Mn^{2+} ions^{58,59}."

Supplementary Fig. 25 | XPS spectra of $\text{Ca}_{1-x}\text{Ga}_4\text{O}_7:x\text{Mn}^{2+}$ ($x = 1 \times 10^{-4}$) before and after photoexcitation. a, $\text{Mn}2p$ spectra. b, $\text{Mn}3s$ spectra.

Although photo-induced oxidation of Mn^{2+} ions has been reported in the literature, these irradiations generally belong to high doses of high-energy rays and high-intensity UV light, e.g., γ -rays (1.6 kGy/h; Ref. R4), X-rays (55 kV, 30 mA, 20 min; Ref. R5), UV irradiation (200 W/cm², 20 min; Ref. R6). In comparison, in our work, the intensity of UV photoexcitation is very low, only 0.25-0.75 W/cm², and the photoexcitation time is very fast, only a few seconds. Such low intensity and fast UV irradiation may not be

sufficient to oxidize Mn^{2+} ions.

According to your suggestion, we have performed EPR tests on the $\text{CaGa}_4\text{O}_7:\text{Mn}^{2+}$ phosphors. However, the EPR spectra (**Supplementary Fig. 27**) show that no characteristic signals of Mn^{2+} and Mn^{4+} ions were detected (Mn^{3+} is normally EPR-silent). We attribute the absence of Mn^{2+} signals (sextet peaks) to the low concentrations of Mn^{2+} in this work (0.01 mol%–0.1 mol%). The ESR spectra reveal a significant increase in the characteristic signal of the singly ionized oxygen vacancies (V_o^+) at $g = 2.003$ upon photoexcitation (**Fig. 7b**). This finding indicates that the doubly ionized oxygen vacancies (V_o^{++}) within the material trap the photoexcited electrons, leading to the formation of the color center. We have also added the investigation on the photochromic properties of undoped CaGa_4O_7 and Mn^{2+} -doped CaGa_4O_7 (**Supplementary Fig. S26**). The results show that CaGa_4O_7 and $\text{CaGa}_4\text{O}_7:\text{Mn}^{2+}$ exhibit similar photochromic phenomena under UV irradiation, and the apparent color of both materials changes significantly from milky white to brown. Moreover, the absorption peaks of the two materials after photoexcitation show the same peak position, which indicates that the photochromic phenomena both originated from the same color centers.

In addition, the PLE spectra show only transitions of Mn^{2+} ions from the ground state ${}^6\text{A}_1({}^6\text{S})$ to the excited states ${}^4\text{A}_1({}^4\text{G})$, ${}^4\text{E}({}^4\text{G})$, ${}^4\text{T}_2({}^4\text{G})$ and ${}^4\text{T}_1({}^4\text{G})$ and do not show any transitions from the defects and high-valent Mn ions (Supplementary Fig. 7).

Supplementary Fig. 27 | EPR spectra of $\text{Ca}_{1-x}\text{Ga}_4\text{O}_7:x\text{Mn}^{2+}$ ($x = 1 \times 10^{-4}$ and 10×10^{-4}). The results show that no characteristic signals of Mn^{2+} ions were detected due to low concentration doping of Mn^{2+} ions and no signals of Mn^{4+} were found.

Figure 7. Optical transition and energy transport mediated by defects. b, ESR spectra of $\text{CaGa}_4\text{O}_7:\text{Mn}^{2+}$ before and after photoexcitation.

Supplementary Fig. 26 | Photochromic coloring and decolorization processes of CaGa_4O_7 and $\text{CaGa}_4\text{O}_7:\text{Mn}^{2+}$. **a,d,** Diffuse reflectance spectra and photochromic photographs of CaGa_4O_7 and $\text{CaGa}_4\text{O}_7:\text{Mn}^{2+}$ disks under 254 nm irradiation for different photoexcitation time (0 s–3 min). **b,e,** Diffuse reflectance spectra and photochromic photographs of CaGa_4O_7 and $\text{CaGa}_4\text{O}_7:\text{Mn}^{2+}$ disks after 254 nm photoexcited (3 min) with different delay times (0–30 min). **c,f,** Absorption spectra and photochromic photographs of CaGa_4O_7 and $\text{CaGa}_4\text{O}_7:\text{Mn}^{2+}$ disks without/with photoexcitation (254 nm, 3 min). The results show that the CaGa_4O_7 and $\text{CaGa}_4\text{O}_7:\text{Mn}^{2+}$ disks exhibit similar photochromic phenomena under UV irradiation, and the apparent color of both materials changes significantly from milky white to brown. Moreover, the absorption peaks of the two materials after photoexcitation show the same peak position, which indicates that the photochromic phenomena of both originates from the same color centers.

References used for responses:

- R4. Zhdachevskii, Ya. et al. Optical observation of the recharging processes of manganese ions in $\text{YAlO}_3:\text{Mn}$ crystals under radiation and thermal treatment. *J. Phys.: Condens. Matter* **18**, 5389–5403 (2006).
- R5. Romet, I. et al. Recombination luminescence and EPR of Mn doped $\text{Li}_2\text{B}_4\text{O}_7$ single crystals. *Opt. Mater.* **70**, 184–193 (2017).
- R6. Shin, H. et al. Photochromic effect in stoichiometric $\text{LiNbO}_3:\text{Fe},\text{Mn}$. *Jpn. J. Appl. Phys.*

5. Density functional theory (DFT) calculations identified defect levels close to the conduction band minimum (CBM) for Si, Al, Zn, and Ge substituting Ga defects. The authors also found a deep occupied level for the Ga-vacancy? It is unclear what V_{GaO} means? A double vacancy or single vacancy of gallium? The authors compared the energy difference between these Kohn-Sham levels and the experimentally observed maximum intensity in the PL spectrum (~ 2.6 eV). For the yellow luminescence, the authors suggest that the Mn substituting Ga defect interacts with the O-vacancy where the former creates defect levels close to CBM whereas the different charge states of O-vacancy create levels close to CBM and deep in the band gap. Unfortunately, the authors do not define whether these defect levels are empty or occupied which makes it difficult to understand the arguments of the authors. Again, the Kohn-Sham level of the positively charged O-vacancy is directly compared to the observed trap level at ~ 2.3 eV.

[Authors' Response]: Thank you very much for your comments. During this revision, we have been invited to join this work and we also agree with your comments on the theoretical calculations in the previous manuscript. We agree with you that previous theoretical calculations have mistakenly and rashly stacked the complicated mechanisms together, which causes confusion for you and other Reviewers. We appreciate your comments and we have reached a consensus with other authors to clarify some points and reformulate some parts of the work. Please kindly find the detailed responses to your concerning questions below.

-The authors also found a deep occupied level for the Ga-vacancy?

Yes, we have identified the deep defect for Ga-vacancy in previous theoretical calculations, which are not well explained. In our updated theoretical calculations, two deep defect levels of Ga-vacancy are identified, which is an empty unoccupied state near the mid-gap. Owing to the limited computational resources, these defect levels are not absolutely correct, which is due to the systematic error by DFT. Our theoretical calculations at least indicate the general physicochemical trend, which facilitates the reformulation of the mechanisms.

- It is unclear what V_{GaO} means? A double vacancy or single vacancy of gallium?

V_{GaO} is a type of Schottky defect in the host material. The Schottky defect is formed by the monovacancies of nearby cations and anions, forming a cavity in the material. During the synthesis of the material, the formation of microcavity is possible, especially during the annealing period, which is induced by electrostatic interaction. In this type of defect, the electrons left by the anion at the Schottky defect site are neutralized by the two holes induced by the cation. In our previous work, the formation of Schottky defects has been found to be possible for oxide-based materials, which also contributes to the luminescence performances [*Phys. Chem. Chem. Phys.*, 2016, 18, 25946-25974; *Phys. Chem. Chem. Phys.*, 2017, 19, 1190-1208]. In the updated theoretical calculations, two types of Schottky defect V_{CaO} and V_{GaO} have been considered.

-The authors compared the energy difference between these Kohn-Sham levels and the experimentally observed maximum intensity in the PL spectrum (~ 2.6 eV). For the yellow luminescence, the authors suggest that the Mn substituting Ga defect interacts with the O-vacancy where the former creates defect levels close to CBM whereas the different charge states of O-vacancy create levels close to CBM and deep in the band gap. Unfortunately, the authors do not define whether these defect levels are empty or

occupied which makes it difficult to understand the arguments of the authors. Again, the Kohn-Sham level of the positively charged O-vacancy is directly compared to the observed trap level at ~2.3 eV.

We highly agree with you that previous theoretical calculations have not supplied very clear explanations for the defect levels. We also think that it is not rigorous to discuss the fine defect levels with direct comparisons with the trap levels characterized by ThL, which have been revised in the updated theoretical calculations. In our previous works, we have also discussed that theoretical calculations cannot directly match with the ThL results [*Phys. Chem. Chem. Phys.*, 2017,19, 9457-9469]. In particular, the trap levels and related defect formations will change as the temperature varies, leading to a widely distributed “band” in the experimentally measured ThL spectra instead of a discrete energy level, which indicates the multi-centers of charge carrier recombination under external thermodynamic stimulations. The broad peak of ThL spectra covers the detailed defect level information, leading to possible errors up to 50% for the direct comparison between defect levels of DFT calculations and trap depth characterized by ThL. In the updated theoretical calculations, DFT calculations have been performed by the K-S minimizations with structure ionization relaxations. For all the calculations, we are focusing on the electronic structures and pathways based on the single particle levels. We have considered two situations for all the intrinsic defect levels in both CaGa₄O₇ and CaGa₄O₇:Mn²⁺. All the electronic structures of intrinsic defects have been demonstrated in the supporting information. Meanwhile, all the defect levels have been summarized in **updated Figure 6e-f**, where occupied and unoccupied states are clearly identified. In the updated theoretical discussions, all the defect levels do not aim to directly match with the PL levels. Instead, we highlight the electronic excitation pathway to reveal the potential electron transfer pathway to realize both blue emissions and yellow emissions for multi-modal luminescence.

Based on the comprehensive screening of all the intrinsic defects, we have summarized all the gap states to identify the possible electron transfer pathway in the host materials (**Fig. 6e**). The intrinsic defects i_O , V_{Ga} , V_{GaO} , and V_O dominate the electron transfer for the blue emission from the defect level to the ground state with photo emissions near 2.60 eV (476 nm). Meanwhile, the electron transfer from i_{Ga^+} to i_{Ga} and $V_{O1^{2+}}$ to V_O also supplies blue emission with wavelengths of 457 nm and 490 nm, respectively. Moreover, the direct electron transfer from the impurity Cu_{Ca} defect levels to the ground states also enables the photoemission around 2.61 eV (475 nm), which slightly enhances the blue emission due to the low concentration of impurity. For the CaGa₄O₇: Mn²⁺, we have also summarized the main defect levels with positions close to the Mn²⁺ based on the PDOSs screening results (**Fig. 6f, Supplementary Fig. 23**). It is noted that the introduction of Mn²⁺ induces much higher defect levels, supplying abundant and flexible electron transfer pathways to achieve dynamic multicolor emissions. There are abundant occupied states induced near the VBM, which reduces the electron transfer barriers to promote the overall luminescence performances. The i_{Ga} and i_O are the main contributions to the shallow traps below CBM and the oxygen vacancies mainly induce deep defect levels near the mid-gap, where the empty states play as the traps for intrinsic blue emissions and promote the tunneling effect.

Fig. 6. Theoretical explorations of intrinsic defect assisted luminescence mechanism. The (a) PDOS and (b) bandstructure of CaGa_4O_7 . The (c) PDOS and (d) bandstructure of $\text{CaGa}_4\text{O}_7:\text{Mn}^{2+}$. The summarized single particle levels in (e) CaGa_4O_7 and (f) $\text{CaGa}_4\text{O}_7:\text{Mn}^{2+}$. Blue line: occupied states; Orange line: unoccupied states. The defect formation energies in (g) CaGa_4O_7 and (h) $\text{CaGa}_4\text{O}_7:\text{Mn}^{2+}$. (i) The formation energy changes of oxygen vacancies with different distances to Mn^{2+} . (j) The formation energy of defects with temperature changes. The mechanism diagram for (k) room-temperature PL, (l) high-temperature PL, and (m) ML.

6. As will be explained below, the comparison of DFT results to experiments stands on very weak feet, so it is questionable whether these DFT results can be applied for the discussion of the excitation and de-excitation processes. In Figs. 6l-n, the authors mix the single particle picture (band diagram) and the many-body levels (e.g., 6A1, 4T1). This is wrong as explained in Refs. [Phys. Rev. B, 104, 235301 (2021), Nanophotonics 12, 359 (2023)].

[Authors' Response]: Thank you very much for your comments. We highly appreciate your comments, and we aim to supply the general and simple physicochemical trend to understand the luminescence mechanism. We sincerely apologize that previous mechanism figures have been mistakenly and rashly stacked with the overcomplicated mechanism together without rigorous considerations, which caused strong confusion to you and other Reviewers. The luminescence mechanism figures in the previous manuscript are proposed based on the combinations of experimental characterizations and DFT results, which are however rashly demonstrated. Currently, there are many different research groups that are working on the qualitative descriptions of the

excitation and de-excitation processes. Based on your comments, **Figure 6 l-n** has been updated and simplified based on the DFT results of single particle level calculations to explain the potential luminescence mechanisms as **updated Figure 6k-m**. In addition, we have cited the two references you mentioned in the comment.

Updated Fig. 6k-m, The mechanism diagram for (k) room-temperature PL, (l) high-temperature PL, and (m) ML.

Accordingly, we have demonstrated the luminescence mechanisms based on different external stimulations based on the single particle level summary (**Fig. 6k-m**). For the room temperature PL, the charge separation process is initiated by the strong photoexcitation, where the electrons are excited from VB to CB, and the holes are left in the VB (**Fig. 6k**). For the $\text{CaGa}_4\text{O}_7: \text{Mn}^{2+}$, there are only limited defects formed during the synthesis, where i_{Ga} and i_{O} supply limited trap states near the CBM. The blue emissions of the host materials are mainly dominated by the defect states of V_{O} and i_{O} near the mid-gap position at 2.5-3.0 eV above the VBM. The yellow emission of Mn^{2+} is mainly induced by the energy transfer from the non-radiative recombination of electrons and holes. Since the defect states of V_{O} are slightly higher than the state of Mn^{2+} , the electron transfer competition results in the suppression of yellow emission, realizing the dynamic color modulations. The high-temperature PL is strongly associated with the newly formed defects under thermal perturbations (**Fig. 6l**). As DFT calculations informed, the interstitial defects and oxygen vacancies will become much easier to form, especially in the range of 500-600 K. Under such a thermal perturbation, more abundant defect states are formed near VBM and the mid-gap, promoting the electron transfer from V_{O} states to the shallow traps i_{Ga} and i_{O} states. This further strengthens the electron release from trap states to the Mn^{2+} states, leading to enhanced yellow emissions with reduced blue emissions. For the ML, the local piezoelectric potential causes a tilt in the band structure, which is revealed by DFT calculations (**Fig. 6m**). Such a band tilt will reduce the trap depth, enabling faster and more efficient electron release from the trap states to Mn^{2+} to produce the yellow luminescence. The presence of deep traps ensures a continuous supply of electrons to shallow traps during the ML. Therefore, cyclic mechanical stimulation applied to ML elastomers yields an intense ML with high mechanical durability.

7. The authors did not define which functional did they apply. This is a red flag against publication to any scientific journal. The authors mention that they applied DFT-1/2 method for self-interaction correction. The validity of this method is restricted. Many-body perturbation methods and wave functions methods are robust methods to heal the potential band gap underestimation of the host material by the applied DFT functional and the failure in the accurate position of the defects' ionization energies. The authors may calculate the charge transition levels in a traditional fashion with applying charge

correction for the charged defect species with a well-chosen functional and/or posteriori correction methods.

[Authors' Response]: Thank you very much for your comments. We fully understand your concerns on the sub-level mistakes of the functional selection, which we also think is not appropriate. If we are the Reviewers, we will also have the same concerns as you. We highly agree with you that the self-interaction correction is significant even for the high-level functionals due to the Koopmans conditions, which is still challenging for current DFT methods. Although many methods have been proposed to solve self-interaction, most of them can only minimize the self-interaction errors rather than complete corrections, and some of them are semi-empirical methods. Researchers have devoted considerable efforts to correcting the self-interaction energy error by treating only the spurious error component within their calculation method/functional. For example, the introduction of hybrid density functional methods, where part of the local DFT exchange is replaced with the statically screened Hartree–Fock (HF) exchange, which maintains the original inter-electron correlation and effectively cancels the spurious self-interaction effects.

In our updated theoretical calculations, we have applied the DFT+U method by CASTEP code [*Zeitschrift Fur Kristallographie*, 2005, 220, 567-570]. To improve the accuracy of the electronic structure calculations, the Hubbard U parameters have been introduced in the Perdew-Burke-Ernzerhof (PBE)+U calculations with a kinetic cutoff energy of 880 eV based on the Anisimov-type rotational invariant DFT+U method [*J. Phys.: Condens. Matter*, 1997, 9, 767]. To minimize the effect of the localized hole states produced by the 2p orbitals of the O sites, the self-consistently determined Hubbard U potentials have also been applied to the O-2p orbitals as many previous works of oxide materials [*Phys. Rev. B: Condens. Matter Mater. Phys.*, 2009, 80, 085202; *Phys. Rev. B: Condens. Matter Mater. Phys.*, 2010, 81, 205209; *J. Phys. Chem. C*, 2010, 114, 2321; *J. Phys. Chem. C*, 2011, 116, 2443]. We chose the (3s, 3p, 3d, 4s) states as the valence states of Ca, (3d, 4s, 4p) for Mn, (4s, 4p) for Ga, and (2s, 2p) for O. The norm-conserving pseudopotential is accordingly generated based on the OPIUM code in the Kleinman-Bylander form to reduce the systematic error induced by the overlapping of the atomic core-valence electron densities [*Phys. Rev. Lett.*, 1982, 48, 1425-1428]. Meanwhile, the RKKJ method has been used for the optimization of pseudopotentials [*Phys. Rev. B: Condens. Matter Mater. Phys.*, 1990, 41, 1227-1230]. To guarantee electronic minimization and convergence, we introduce the ensemble DFT (EDFT) method to solve the Kohn-Sham equation [*Phys. Rev. Lett.*, 1997, 79, 1337]. The reciprocal space integration was performed by *k*-point sampling with a grid of $10 \times 10 \times 10$ *k* points for the CaGa₄O₇ unit cell, and a grid of $3 \times 3 \times 5$ for defect electronic structure calculations in the supercells. The convergence criteria have been set as follows: 1) the total energy should be smaller than 5.0×10^{-7} eV per atom; the Hellmann–Feynman force should be lower than 0.01 eV Å⁻¹ per atom. With our self-consistent determination process [*Inorg. Chem.*, 2015, 54, 11423; *Phys. Chem. Chem. Phys.*, 2016, 18, 13564; *J. Phys. Chem. C*, 2014, 118, 24248; *Philos. Mag.*, 2014, 94, 3052; *Solid State Commun.*, 2016, 230, 49; *Solid State Commun.*, 2016, 237–238, 34], the on-site Hubbard U parameters are 2.513, 5.013 and 2.700 eV for Ca, Ga and Mn, respectively.

Based on our updated theoretical work, we have applied the DFT+U method, which adds an on-site repulsive potential U for the localized d orbitals of solids to cancel the self-interaction energy. This strategy tries to minimize the self-interaction energy errors

from another perspective. However, the choice of the U parameter for the correction to match the experimentally reported data is primarily semi-empirical and still leads to errors in band structure calculations, especially in defect-state calculations. In fact, defect-state levels in the band structure arise from the host in the BZ and the area correlated with the excited states of the band structure. Although our current calculation method that closely describes the band structure may not be adequate to correctly reflect the defect states, especially the deep-level states, we have tried our best to supply more reliable results than previous calculations. More importantly, our theoretical calculations have supplied a general and simple physicochemical trend to understand the luminescence mechanisms. Since we have limited computational resources, it is too difficult for us to choose the high-level functionals, which usually require a very long period of calculations based on ultra-high computational power and loadings. Therefore, we think that our updated theoretical calculations are sufficient to supply references to the experimental results.

8. In the excitation process, the electron-hole interaction will determine the final excitation energy (e.g., can be calculated by Bethe-Salpeter equation) and the energy difference between Kohn-Sham levels is a rather poor estimation for the excitation energy. Furthermore, if two defects are involved in the excitation then the electron-hole interaction energy depends on the distance between the donor and acceptor defects. The authors did not discuss this issue at all. In addition, the maximum of the PL signal may be associated with the phonon participation in the luminescence process which is again completely ignored by the authors (this effect can be very large in these materials). The overall impression about the application of DFT theory is that the authors do not understand deep enough the defect physics at all, and they oversimplify the picture to find some supportive data for their experimental findings. In fact, the quality of the DFT studies and the validation of the results are very far from the point where a direct comparison to the experimental results can be reliably done.

[Authors' Response]: Thanks very much for your comments. We would like to clarify that the excitation energy in previous theoretical calculations does not represent the energy of the first excited state in DFT calculations. We apologize that the previous descriptions have some discrepancies that cause your confusion. To obtain the excitation energy as you suggested, we have used different methods with embedded B-S codes on the compilation platform to do the initial tests including the Intel Fortran with MKL math lab and open source GFortran with Atlas Math lab based on Intel MPI library. However, the first cycle minimization to obtain the ground state wavefunction already costs a week. The final completion requires around 4169 cycles, which leads to the untra-long computation period. We sincerely apologize that our current computation power and resources cannot support such calculations. The Kohn-Sham levels are not suitable for the excitation energy since it only considers the ground state energy. The excitation energy should be resolved by the EDDFT, which usually has been applied to small molecules. Although the EDDFT can also be applicable to the solids, the convergence becomes extremely difficult, which shows a strong dispersion effect on the excitation energy calculations.

In the updated theoretical calculations, we have supplied discussions of the electronic excitation pathway for photon emissions based on the static calculations of the defect levels. We have considered both intrinsic defects and intrinsic defects with close distance to Mn^{2+} . We have noticed that the close distance between intrinsic defects and Mn^{2+} has resulted in different electronic structures of trap levels. The close distance

between intrinsic defects and Mn^{2+} induces more abundant levels, which allows dynamic electron transfer pathways to modulate and realize the multicolor luminescence. Moreover, most of the defects are relatively close ($< 5 \text{ \AA}$) in the lattice to initiate the electron-hole interactions.

For the participation of phonons in luminescence, we have found that most of the highly impactful or authoritative research works have not considered such an effect [*Nature Commun.*, 2021, 12, 2022; *Nature Commun.*, 2022, 13, 4438; *Nature Commun.*, 2022, 13, 7589]. For current luminescence investigations, we have conducted a literature review and identified that phonon involvements have been rarely considered for the oxide-based hosts. Most of the phonon involvements are applied in the halide perovskites based on the self-trapping states to explain the luminescence properties [*Nature Commun.*, 2023, 14, 234; *Nature Commun.*, 2020, 11, 2344; *Nature Commun.*, 2022, 13, 7589]. The Reviewer is correct that phonon may contribute to the luminescence process. However, in this work, the luminescence process involves the multi-phonon processes, which cannot be quantitatively described even for the TDDFT or electron-phonon coupling calculations by DFT. The phonon dispersion calculated by the DFT also cannot match with the luminescence, which shows very limited correlations. In this work, DFT calculations only aim to supply the general physicochemical trend to demonstrate the potential electron transfer pathways. Moreover, our current experimental characterizations have not found very solid or clear evidence of phonon involvement in luminescence. Therefore, we think that the phonon may have some contributions, but it cannot be quantitatively described in the current work. For this concern, there are time-dependent perturbation DFT for the energy of the first excited state, the electron-phonon coupling coefficient, and the excitation in the semiconductors. We are not very clear which is the Reviewer's opinions. Moreover, the DFT calculations of defect levels are static while the phonons are dynamic, which leads to the electron-phonon coupling coefficient. In the meantime, the TDDFT calculation of excitation energy is another different type of strategy.

In our revised manuscript, we have comprehensively investigated the contributions of different defect levels to the luminescence properties. With systematic investigations of all the possible defects, the DFT calculations have supplied important insights into understanding the underlying mechanisms for the multi-color luminescence properties of $\text{CaGa}_4\text{O}_7:\text{Mn}^{2+}$. Our theoretical calculations have offered a good physicochemical trend to understand the potential origins of luminescence, which are also highly critical for experimental groups in designing novel and advanced luminescence materials.

10. Minor issues:

Line 146: 'x' is missing in the composition

[Authors' Response]: Thanks very much for your careful checking and kind reminder. We apologize for the typo. After a careful check, we have updated the expression in the revised manuscript.

Reviewer #3: I co-reviewed this manuscript with one of the reviewers who provided the listed reports. This is part of the Nature Communications initiative to facilitate training in peer review and to provide appropriate recognition for Early Career Researchers who co-review manuscripts.

[Authors' Response]: Thank you very much for your co-review and valuable comments. We have carefully considered the reviewers' suggestions and have made revisions accordingly. We hope that our responses and revisions meet the high standards you expect for publication.

In the end, we would like to thank for the precious time of the reviewers and editors. We sincerely hope that our point-to-point response and the revised manuscript can address your concerns and satisfy your requirements for publication. We would be grateful if we have the chance to share our work with readers of *Nature Communications*.

Reviewers' Comments:

Reviewer #2:

Remarks to the Author:

The manuscript titled "Enabling Dynamic Multicolor Emission and Photo-Thermo-Mechanically Responsive Luminescence via Trace Mn²⁺ Doping in Self-Activated CaGa₄O₇" explores the optical properties of Mn-doped gallate crystals. The authors reveal that these crystals display distinct time-dependent dual luminescence, temperature-dependent emission, and mechanoluminescence, which could be beneficial for the detector and anti-counterfeiting industry.

The revised manuscript has improved readability and a more detailed and comprehensive presentation of the results. The effort put in to address the reviewer's questions and make the necessary modifications has eliminated any doubt about the data.

The quality of the manuscript has been significantly improved with the executed modifications and the updated theoretical calculation.

I recommend that it be published.

Response to Reviewers

Reviewer #2 (Remarks to the Author):

The manuscript titled "Enabling Dynamic Multicolor Emission and Photo-Thermo-Mechanically Responsive Luminescence via Trace Mn²⁺ Doping in Self-Activated CaGa₄O₇" explores the optical properties of Mn-doped gallate crystals. The authors reveal that these crystals display distinct time-dependent dual luminescence, temperature-dependent emission, and mechanoluminescence, which could be beneficial for the detector and anti-counterfeiting industry.

The revised manuscript has improved readability and a more detailed and comprehensive presentation of the results. The effort put in to address the reviewer's questions and make the necessary modifications has eliminated any doubt about the data.

The quality of the manuscript has been significantly improved with the executed modifications and the updated theoretical calculation.

I recommend that it be published.

[Authors' Response]: Thank you very much for your affirmation of our work, as well as your positive comments and valuable suggestions on the manuscript. Thanks very much for your recommendation of publication.